# Electrochromic semiconductors as colorimetric SERS substrates with high reproducibility and renewability

Shan Cong[1,2], Zhen Wang[1], Wenbin Gong[1], Zhigang Chen[1], Weibang Lu[1], John R. Lombardi[3] & Zhigang Zhao[1,4]

Electrochromic technology has been actively researched for displays, adjustable mirrors, smart windows, and other cutting-edge applications. However, it has never been proposed to overcome the critical problems in the field of surface-enhanced Raman scattering (SERS). Herein, we demonstrate a generic electrochromic strategy for ensuring the reproducibility and renewability of SERS substrates, which are both scientifically and technically important due to the great need for quantitative analysis, standardized production and low cost in SERS. This color-changing strategy is based on a unique quantitative relationship between the SERS signal amplification and the coloration degree within a certain range, in which the SERS activity of the substrate can be effectively inferred by judging the degree of color change. Our results may provide a first step toward the rational design of electrochromic SERS substrates with a high sensitivity, reproducibility, and renewability.

[1] Key Lab of Nanodevices and Applications, Suzhou Institute of Nano-Tech and Nano-Bionics, Chinese Academy of Sciences (CAS), 215123 Suzhou, China. [2] Key Laboratory of Multifunctional Nanomaterials and Smart Systems, Chinese Academy of Sciences (CAS), 215123 Suzhou, China. [3] Department of Chemistry, The City College of New York, New York, NY 10031, USA. [4] Division of Nanomaterials, Suzhou Institute of Nano-Tech and Nano-Bionics, Chinese Academy of Sciences, 330200 Nanchang, China. Correspondence and requests for materials should be addressed to Z.Z. (email: zgzhao2011@sinano.ac.cn)

Surface-enhanced Raman scattering (SERS) is a powerful spectroscopic tool, characterized by rapid, nondestructive, ultra-sensitive and fingerprint diagnostics, so it has a wide spectrum of promising applications in chemical and biological analysis at trace levels. With the indispensable use of substrate materials, SERS can give dramatically enhanced Raman signals for analytes with small Raman scattering cross-sections, which are usually difficult to detect by traditional Raman spectroscopy[1,2]. Typically, besides the magnitude of enhancement, uniformity, reproducibility, and renewability are additional important criteria in the practical evaluation of SERS substrates[3,4]. Among them, uniformity is defined as the absence of diversity or variation in Raman signal intensity from spot to spot within a single substrate, while reproducibility refers to test results in different batches of SERS substrates. In previous studies, particular emphasis has been given to the search for SERS substrates with a high level of SERS enhancement and uniformity. In contrast, the reproducibility and renewability of SERS substrates have been much less explored, although they are also both scientifically and technically important due to the great need for quantitative analysis, standardized production and low cost, and constitute one of the major scientific problems in the SERS field. If the problems could be solved, it would allow for quantitative and qualitative SERS analysis in practice. However, constructing SERS substrates with excellent reproducibility and renewability remains a great challenge. For metallic SERS substrates, that is, Cu, Ag, and Au, plasmonic hot spots have been proved to decisively contribute to SERS, where the electromagnetic field is greatly concentrated[5]. Unfortunately, these hot spots are randomly distributed, hard to exactly reproduce, and display considerable uncertainties, which is one reason for the poor reproducibility of the SERS signals of metallic SERS substrates. In fact, it was recently demonstrated that the hottest SERS-active sites account for only 63 in every 1,000,000 sites, but contribute 24% of the overall SERS intensity[6]. Controlling fabrication to ensure a reasonable distribution of the hottest SERS-active sites throughout a metallic substrate surface, allowing both highly sensitive and reproducible Raman signals, is still technically unfulfillable, not to mention the high cost of processing nanostructures with coinage metals. In addition, the traditional metallic substrates for SERS are usually single-use and non-renewable, which has further added to the cost and greatly limited their applications.

In contrast, in the case of semiconductor substrates, the enhanced Raman signals are considered to derive from other resonant contributions such as charge-transfer, excitonic, and molecular resonance, mostly due to the electronic transitions between the analytes and semiconductor substrate[7]. Such a chemical interaction-based mechanism offers new opportunities for the simultaneous fulfillment of important criteria for consideration in SERS measurements using semiconductor SERS substrates. This, in turn, calls for new, very different strategies for SERS substrate design. For example, in terms of SERS signal amplification, recent studies demonstrate that the structural modulation of semiconductor materials, that is, oxygen defects or crystal facet engineering, is an effective means of creating SERS-active semiconductor substrates due to the altered electronic structures of semiconductor materials[8–14]. However, these strategies still neglect other critical issues of semiconductor SERS substrates, such as uniformity, reproducibility, and renewability, largely due to the difficulty in limiting and controlling the spatial and temporal variability of the surface structure, chemical composition, and electronic distribution of semiconductor materials using current treatments. Meanwhile, direct colorimetric detection of the SERS activity of semiconductors is believed to be a particularly attractive feature for SERS systems, because it would allow rapid preliminary discrimination of SERS activity through a color change observable by the naked eye. However, to the best of our knowledge, it has never been attempted. Therefore, developing semiconductor SERS substrates with high colorimetric functionality, reproducibility, and renewability could potentially lead to a breakthrough in semiconductor SERS in terms of quantitative analysis, standardized production, and low cost. Nonetheless, it remains a significant challenge.

Using tungsten oxide as a model material, we propose a strategy to simultaneously ensure the signal amplification, uniformity, reproducibility, and renewability of semiconductor SERS substrates. We incorporate colorimetric functionality into semiconductor SERS substrates by a color-changing electrochromic operation that can be described as a reversible reduction/oxidation process accompanying the insertion/extraction of ions and electrons in a quantitative way. Impressively, upon charging, the deep-blue-colored substrate shows a striking increase of SERS enhancement compared with the uncolored substrate towards rhodamine 6G (R6G), crystal violet (CV), and Victoria blue B (VBB), when the excitation wavelength is fixed to be 532 nm (the physicochemical properties of the analytes are detailedly outlined in Supplementary Note 1). More importantly, a clear quantitative relationship can be found between the SERS enhancement of the colored substrate and the amount of intercalated charges by systematically varying the negative voltage, which enables a controlled modulation of the chemical and electronic structures of semiconductor SERS substrates and in turn affects their SERS performances. Additionally, due to this quantitative relationship, information on the SERS activity of a semiconductor SERS substrate can be effectively acquired by judging the degree of color change. The developed approach is also applicable to other semiconductors (e.g., nickel oxide (NiO)) and other intercalated cations (such as $H^+$, $Li^+$, $Na^+$, $K^+$, $Mg^{2+}$, and $Al^{3+}$), thus demonstrating the universality and extendibility of this method.

## Results

**Preparation and characterization of electrochromic SERS substrates**. Taking tungsten oxide as an example, the electrochromic SERS substrates are first grown on fluorine-doped tin oxide (FTO) glass using a direct current (DC) magnetron sputtering technique at room temperature, which is widely used for the large-scale preparation of thin films. The as-sputtered films are then charged and discharged potentiostatically at different voltages with various cations including $H^+$, $Li^+$, $Na^+$, $K^+$, $Mg^{2+}$, and $Al^{3+}$ in aqueous solution for reversible electrochromic coloration/decoloration (see details in Methods). After the electrochromic treatments, the different colored and uncolored substrates are used for SERS measurements.

The resulting pristine uncolored substrates are comprehensively characterized by scanning electron microscopy (SEM), X-ray diffraction (XRD), Raman spectroscopy, and X-ray photoelectron spectroscopy (XPS), to study their morphologies, chemical bonding, phase structures, and other properties. Morphologically, the pristine uncolored film on FTO glass shows a rather uniform and dense surface in the top-view SEM images, which is constructed by closely packed coral-like structures in the size range of 200–400 nm (Fig. 1a). The thickness of the sputtered film over FTO glass is estimated to be around 300 nm, as measured from the cross-sectional SEM image (Fig. 1b). In comparison with other semiconductor SERS substrates prepared from suspended powders (such as $W_{18}O_{49}$ sea-urchin-like nanowires), the uniform coverage of the substrate by the dense film, with low variation in the surface roughness, as shown by the AFM measurements (Supplementary Fig. 1), may benefit the uniformity of the SERS signal. The structural identification of the pristine uncolored film is first performed by XRD analysis, in

which two broad and overlapping peaks with intensity maxima at 25° and at 55° are clearly visible, implying the amorphous nature of the pristine film (Supplementary Fig. 2a). Raman analysis further verifies its amorphous nature (Supplementary Fig. 2b), as the characteristic bands that would indicate the formation of crystalline tungsten oxide, including the O-W-O vibration (717 $cm^{-1}$) and W-O-W stretching (807 $cm^{-1}$) (Supplementary Fig. 3), are not observed in the spectrum. Instead, a strong band assigned to the W=O stretching vibration mode can be found at 950 $cm^{-1}$, ascribed to the abundance of terminal O atoms in the amorphous structure[15]. Next, XPS analysis is performed to identify the mixed chemical states of the pristine film. The XPS survey spectrum clearly suggests that the film mainly consists of only two elements: W and O (Supplementary Fig. 4). Specifically, the W 4$f$ core-level XPS spectrum can be deconvoluted into two doublets ($W^{6+}$, $W^{5+}$) with atomic percentages of $W^{5+}$ and $W^{6+}$ fitted to be 11.6% and 88.4%, indicating the presence of mixed tungsten states in the film (Supplementary Fig. 5).

Taking Al ion intercalation as an example, the blue-colored films can be prepared as SERS substrates through electrochemical charging by placing the pristine tungsten oxide films as a working electrode in a three-electrode cell setup (with Pt wire as the counter electrode, Ag/AgCl as the reference electrode, and $AlCl_3$ aqueous solution as electrolyte) (see Methods). It should be emphasized that the chemical composition of the colored films can be temporally and spatially controlled to a large extent by the quantitative intercalation of $Al^{3+}$ ions. For example, the spatial uniformity of the colored films is revealed by SEM and energy-dispersive spectroscopy (EDS) analysis. As shown in Fig. 1c, an almost flat surface morphology for the colored films can be clearly observed in the SEM image, which is nearly unchanged compared with the pristine tungsten oxide films. EDS elemental mapping further confirms the uniform distribution of Al, W, and O in the colored films, satisfying a prerequisite for the homogeneous distribution of SERS-active sites in the colored films (Fig. 1c). Meanwhile, direct verification of the temporal reversibility and batch-to-batch reproducibility of the chemical composition of the colored films can also be inferred from the cyclic voltammograms (CVs), ultraviolet–visible (UV–Vis), and XPS results. Figure 1d illustrates the typical CV curves of the colored films, showing a broad anodic peak around −0.1 V associated with the redox reactions between $W^{6+}$ and $W^{5+}$ accompanied by $Al^{3+}$ intercalation/extraction into the lattice of tungsten oxide. It is clearly shown that after 50 cycles of coloring/bleaching operation, the CV curves are substantially unchanged, implying good reversibility between the coloring and bleaching states. UV–Vis spectroscopy is also used to investigate the temporal change in the composition of the colored films by evaluating their optical properties. As depicted by the change in absorption (monitored at 532 nm) against voltages alternating between −0.5 and 0.2 V, the film changes color repeatedly between the bleached (transparent) and colored (blue) states (Fig. 1e, Supplementary Fig. 6). One can see that the changes in absorption are almost identical in each cycle, further suggesting the reversibility of the composition change. Such a temporally controlled ion intercalation also ensures high batch-to-batch reproducibility of the colored films. As depicted in Fig. 1f, five colored samples prepared in parallel using the same synthetic conditions show almost identical absorption characteristics, indicating good batch-to-batch reproducibility. XPS measurements give further evidence of control over the chemical composition of colored films in different batches (Supplementary Fig. 7). It can be seen that different batches of colored films produced using the same synthetic conditions have similar atomic ratios of $W^{5+}/W^{6+}$, indicative of the reproducibility of chemical composition among batches (Fig. 1g). Such spatial and temporal control of composition is

expected to enable the control of the electronic structures of the colored films, thereby providing a reliable foothold toward fine control of their physical and chemical properties, including SERS properties.

**Quantitative correlation between SERS and coloration degree**. Interestingly, the colored films exhibit much stronger SERS signals than the uncolored films, and even show a unique quantitative relationship between the SERS signal amplification and the coloration degree within a certain range. As shown in Fig. 2a, the pristine uncolored film exhibits a rather weak SERS intensity probably due to rather weak charger-transfer (CT) enhancement when using R6G as a Raman probe ($10^{-4}$ M, molecular structure provided in Supplementary Fig. 8). In contrast, all six colored films obtained at various negative potentials (−0.1, −0.2, −0.3, −0.4, −0.5, and −0.6 V), denoted as Al-1, Al-2, Al-3, Al-4, Al-5, and Al-6, respectively, display much stronger SERS signals of R6G molecules (Fig. 2a). Several characteristic bands of R6G molecules are easily visible at 612, 773, 1360, and 1650 $cm^{-1}$ in these six samples, which can be assigned to in-plane and out-of-plane bending motions of carbon and hydrogen atoms of the xanthene skeleton, and aromatic C–C stretching vibration modes, respectively[16]. Another point worthy of note is that with the increase in the coloration degree of the electrochromic SERS substrates as depicted by analysis of the absorbance at 532 nm (Supplementary Fig. 9), a further increase in the intensity of the SERS signal could be observed (Fig. 2a). To provide a quantitative view for the effect of electrochromic coloration on SERS activity, the enhancement factors (EFs) for different samples are calculated based on the intensity magnification of the most strongly enhanced band (at 612 $cm^{-1}$) compared with that on bare substrate (Supplementary Methods). Impressively, when plotted vs. the absorbance, the EF is found to increase almost linearly up to $2.66 \times 10^4$, reaching a maximum within the absorbance range of 0.3–1.2, nearly 28 times greater than that of the pristine uncolored film (Fig. 2b). Importantly, the obtained quantitative relationship may allow figure ground discrimination of SERS activity in the colored films by the naked eye, because a high degree of electrochromic coloration corresponds to high levels of SERS activity. Furthermore, SERS measurements are also performed for the Al-6 sample under different concentrations of R6G molecules, decreasing from $10^{-4}$, $10^{-5}$, to $10^{-6}$ M (Fig. 2c). It is found that the SERS signals are still conspicuous even when the concentration of R6G solution is decreased to $10^{-6}$ M, suggesting the low detection limit for R6G.

Strikingly, our approach to the activation of SERS activity by electrochromic coloration is also applicable to other cation species such as $H^+$, $Li^+$, $Na^+$, $K^+$, and $Mg^{2+}$. As a result, greatly improved SERS performances are observed for these colored tungsten oxide films when intercalated with all five of the above cations, compared with that of the pristine uncolored film, as shown in Fig. 2d. The rank order of SERS activity for these colored films is $Li^+ > Mg^{2+} > Na^+ > K^+ > Al^{3+} > H^+$. Evidently, the $Li^+$ ion-intercalated film produces the largest SERS enhancement, with the increased EF reaching a maximum value of $8.86 \times 10^4$ for R6G ($10^{-4}$ M) detection. By contrast, the advantage of using trivalent $Al^{3+}$ as the insertion ion is that the colored film intercalated with $Al^{3+}$ is more stable in the open air than its counterparts intercalated with other cations[17], making it especially suitable for the acquirement of reproducible SERS signals. Further investigation demonstrates that the good universality of our approach extends to other electrochromic semiconductors such as NiO (Supplementary Fig. 10). All six NiO films electrochemically activated by intercalation with different cations ($H^+$, $Li^+$, $Na^+$, $K^+$, $Mg^{2+}$, and $Al^{3+}$) are found to

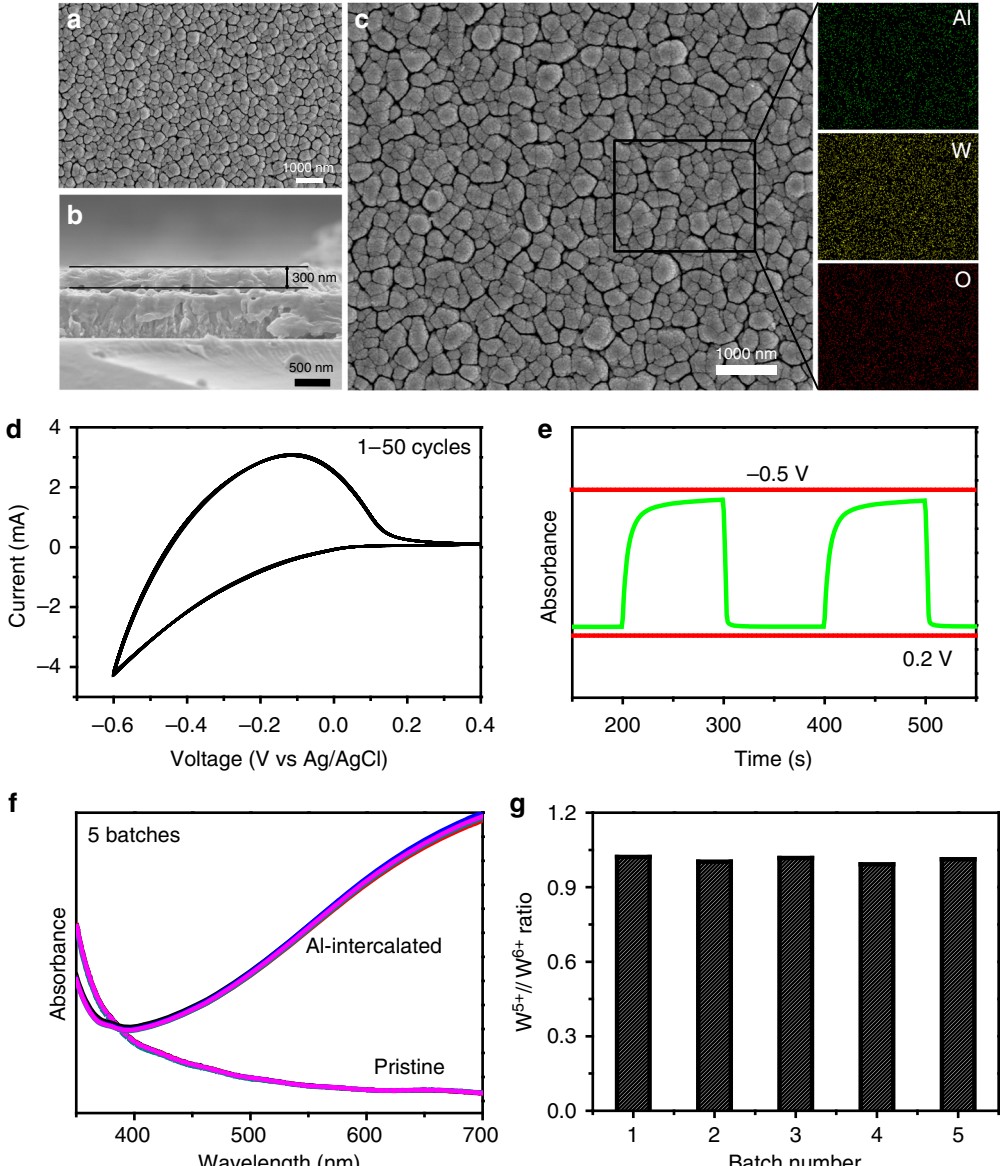

**Fig. 1** Composition control of tungsten oxide through Al intercalation. **a** Top-view and **b** cross-sectional scanning electron microscopy (SEM) images for the pristine tungsten oxide film on fluorine-doped tin oxide (FTO) glass. **c** SEM image of the Al-intercalated tungsten oxide film, and the elemental mapping results of the selected area. **d** Cyclic voltammogram (CV) profiles of the tungsten oxide film for 50 cycles, obtained in 1 M AlCl$_3$ aqueous analyte at a scan rate of 10 mV/s. The 50 curves are overlapping, suggesting good reversibility. **e** Absorbance switching monitored at 532 nm for tungsten oxide, with alternating biases switching between −0.5 and 0.2 V in 1 M AlCl$_3$ aqueous analyte. **f** Absorbance spectra taken for tungsten oxide films from five batches, and those after discharging potentiostatically at −0.5 V for 180 s in 1 M AlCl$_3$ aqueous analyte. **g** Ratio of W$^{5+}$/W$^{6+}$ calculated from deconvolution of X-ray photoelectron spectroscopy (XPS) W 4$f$ core-level spectra for Al-intercalated film in five batches

demonstrate much stronger SERS enhancement compared with the pristine NiO film, further corroborating the effectiveness of electrochromic coloration for increasing Raman detection sensitivity. Despite the similar mechanism of SERS signal augmentation, the ion-intercalated NiO films give a different rank order for the SERS activity: Al$^{3+}$>Mg$^{2+}$>H$^+$>Li$^+$>Na$^+$>K$^+$. The difference is probably linked to the differences in both crystallographic and electronic structures between the two distinct electrochromic materials, which affect the transportation and accommodation of the various inserted cations.

Further investigation demonstrates that the electrochromic SERS substrates are also effective for other types of analytes such as CV and VBB with 532-nm laser excitation (Supplementary Fig. 11). Furthermore, in the case that the laser excitation is not resonant with the molecular absorbance of R6G, the enhanced

SERS signals from R6G are still distinguishable under 633-nm laser excitation (Supplementary Fig. 12), suggesting the generality of this application of the electrochromic SERS substrates. It should be noted that the Raman signals for CV and VBB are relatively weaker as the excitation wavelength (532 nm) is not in direct resonance with the absorption of CV at about 590 nm or VBB around 599 nm. In contrast, the molecular transition between the HOMO and LUMO levels of R6G at 2.3 eV is energetically near that of the excitation laser (532 nm), which provides an important resonant pathway for the observed SERS enhancement. In fact, such thermodynamically feasible resonance can contribute to the overall Raman enhancement through intensity borrowing based on the Herzberg–Teller vibronic coupling, according to the theory established by Lombardi et al[7]. On coupling, the relatively weak charge-transfer resonance

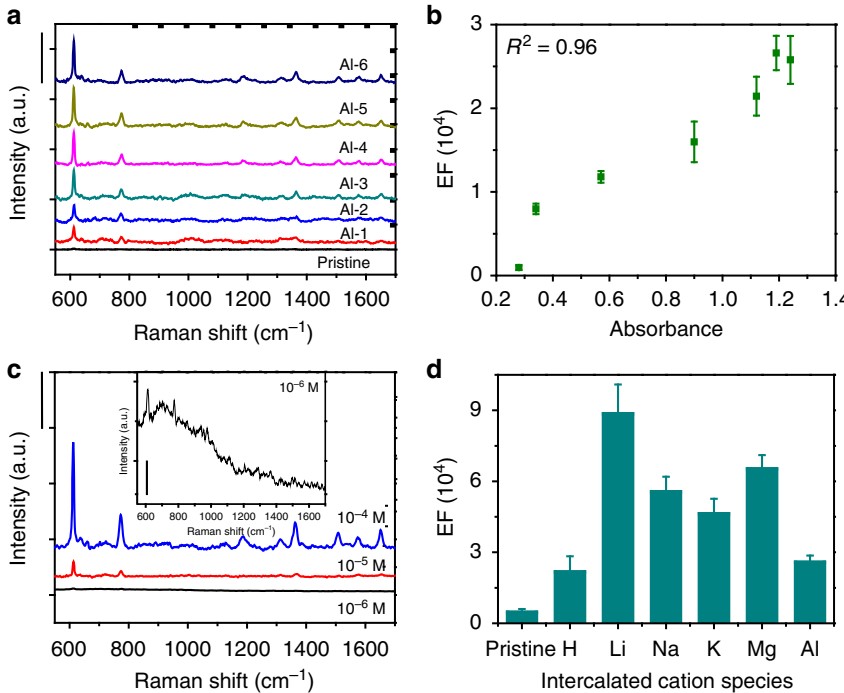

**Fig. 2** Colored tungsten oxide film as surface-enhanced Raman scattering (SERS) substrate. **a** Raman intensity of rhodamine 6G (R6G) ($10^{-4}$ M) on the pristine uncolored film and colored substrates (scale bar: 1000 cps). **b** Statistical evolution of enhancement factor (EF) vs. optical absorbance for pristine and Al-intercalated substrates, indicating a roughly linear increase. A high coefficient of determination ($R^2$) of 0.96 is obtained, demonstrating the linear dependence between EF and optical absorbance. Colored films obtained via Al intercalation at various negative potentials (−0.1, −0.2, −0.3, −0.4, −0.5, and −0.6 V) are denoted as Al-1, Al-2, Al-3, Al-4, Al-5, and Al-6, respectively. **c** Raman spectra of R6G collected for Al-6 substrate at different concentrations of $10^{-4}$, $10^{-5}$, and $10^{-6}$ M (scale bar: 500 cps). Inset: with narrowed y scale for $10^{-6}$ M (scale bar: 10 cps). **d** EF values for pristine and ion-intercalated substrates, which were prepared by discharging potentiostatically at −0.5 V for 180 s in 1 M aqueous analyte of each cation species. Error bars represent means ± SD of the EFs

borrows intensity from the stronger nearby resonances, such as molecular and exciton transitions, as can be expressed by Herzberg–Teller coupling terms. Accordingly, the so-called surface-enhanced resonance Raman scattering (SERRS) effect could be recognized in the R6G/tungsten oxide system, basing on the coincidence of incident photon energy and electronic transition in R6G molecules.

**High uniformity, reproducibility, and renewability**. The quantitative relationship of the coloration degree to the SERS activity of electrochromic SERS substrates, along with good spatial and temporal control of chemical composition, paves the way to achieve high SERS uniformity, batch-to-batch reproducibility, and renewability, which are critically needed for high-performance SERS substrates. First, to examine the SERS uniformity of the colored films, the SERS contours are plotted by line mapping (Fig. 3a), which is conducted from spot to spot on the same substrate for a $1 \times 10^{-4}$ M R6G solution. Among all of the 100 tested spots, each exhibits a favorable capability of enhancing the Raman signal of the R6G molecules. The RSD of the SERS peak at around 612 cm$^{-1}$ is used to further assess the uniformity of the SERS signals on each substrate. The value of the spot-to-spot RSD is calculated to be 7.86%, indicating that the structure and SERS properties of our colored SERS substrate are relatively uniform (Supplementary Fig. 13). Second, one of the greatest hindrances to progress in realizing quantitative SERS analysis is batch-to-batch reproducibility. In fact, for most SERS substrates, the SERS enhancement greatly varies from batch to batch even if the substrate is uniform to a very high degree, depending critically on the substrate preparation. Here we note that deeper colors

represent higher intensities of the SERS signals for our colored SERS substrates, while lighter colors represent lower intensities. The strong quantitative correlation between SERS signal and coloration degree inspires us to find a colorimetric method to achieve excellent batch-to-batch reproducibility. If the degree of coloration of different batches of SERS substrates could be controlled to be uniform, similar SERS activities would be expected. Following this principle, 50 batches of tungsten oxide substrates are colored to the same degree (duplicate spectra at the same intensity and line shape) by means of electrochemical insertion of Al$^{3+}$ ions at −0.5 V. Figure 3b shows the SERS spectra of $10^{-4}$ M R6G collected from 500 randomly selected positions on the 50 batches of colored SERS substrates under identical experimental conditions. These spectrum signals overlap to form a shaded area as outlined in red, and the narrowness of the shaded area indicates that the fluctuation of SERS signals between batches is very small. Accordingly, the batch-to-batch RSD of the vibration at 612 cm$^{-1}$ is calculated to have a record low value of 6.79%, much lower than that of hydrogen-treated samples (30.2%) (Supplementary Fig. 14) and even beyond commercially available Klarite substrates (29.0%) (Supplementary Fig. 15), further demonstrating the excellent batch-to-batch reproducibility. Third, from the economical viewpoint, renewability is another desirable feature of high-performance SERS substrate. However, only a few studies have reported renewable SERS substrates, capable of being used for more than one round of detection[18–20]. One approach to this goal has been photocatalytic treatment. Impressively, our colored SERS substrates provide an efficient and effective avenue for excellent renewability. Based on the idea of colorimetric determination, the tungsten oxide SERS substrates (Al-5) undergo five cycles of coloring-to-decoloring reversible reactions to obtain

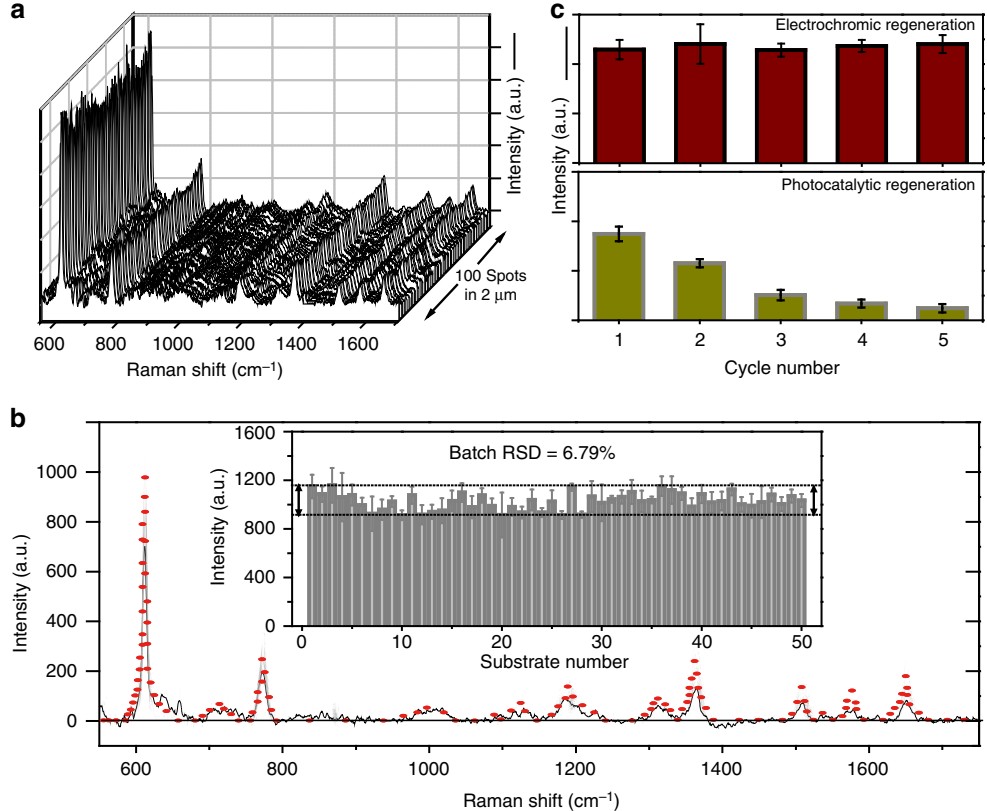

**Fig. 3** High reproducibility and renewability. **a** Surface-enhanced Raman scattering (SERS) spectra in the line-scan across a randomly selected area on the surface of single Al-5 substrate, taken with a 20 nm step size (scale bar: 200 cps). **b** SERS spectra of $10^{-4}$ M rhodamine 6G (R6G) collected from 500 randomly selected positions on 50 batches of Al-5 substrates, with the overlapped signals forming a shaded area as outlined in red. Inset shows the batch-to-batch Raman intensity variation for the 612 cm$^{-1}$ vibrational band measured for these 50 batches of Al-5 substrates. **c** Comparison between SERS intensity of the 612 cm$^{-1}$ vibrational band of single substrates used for multiple cycles, renewed either through electrochromic or photocatalytic methods (scale bar: 400 cps). Error bars represent means ± SD of the SERS intensity

regenerated SERS substrates with restoration of the same coloration degree. The collected SERS spectra from the regenerated SERS substrates upon different coloring–decoloring cycles are compared with those obtained from the freshly prepared substrate (Supplementary Fig. 16). It is noted that neither a large shift in the major Raman peaks nor a significant change in Raman intensity occurs, revealing good renewability. The value of the run-to-run RSD for the vibration at 612 cm$^{-1}$ is 2.52%, which also strongly proves the excellent renewability of the colored SERS substrates prepared by electrochromic coloration (Fig. 3c). The advantage of this colorimetric method in terms of renewability is further revealed by comparison with other control methods such as photocatalytic treatment, which have been previously documented to improve the renewability of SERS[18–20]. As also shown in Fig. 3c, the tungsten oxide substrates (which are oxygen deficient) can clean themselves under UV irradiation by photocatalytic degradation of target molecules adsorbed to the substrates, so that renewability can be achieved. However, it is observed that the EF values continuously decrease from 1.87 × $10^4$ to 0.26 × $10^4$ after five cycles of photocatalytic treatment, leading to the poor renewability of the SERS sensing platform. The main reasons for the large decline in SERS activities may be related to the inevitable accumulation of intermediate products and loss of oxygen vacancies during photocatalytic oxidation. The efficacy of our electrochromic renewability scheme is also examined by two other means (UV–Vis and Raman spectroscopies) (Supplementary Fig. 17). It could be observed that the spectroscopic profiles obtained from the regenerated SERS substrate after electrochromic coloration are almost identical to those

obtained from the freshly prepared substrate, along with the absence of clearly discerned signals ascribed to R6G analyte, indicating good renewability that makes the surface able to recover.

**Structural mechanism underlying electrochromic SERS.** Unlike conventional plasmonic metallic SERS substrates that produce the SERS effect using controllable formation of hot spots, semiconductor SERS substrates enable SERS enhancements through vibronic coupling of several resonances such as charge transfer, excitonic, and molecular resonances, in which the electronic structures play crucial roles in determining the SERS performances[7]. Hence, we explore here the composition-dependent electronic structures of electrochromic SERS substrates under different coloration conditions to clarify the origin of high SERS reproducibility and renewability.

When ionic species such as Al$^{3+}$ are inserted into the lattice of tungsten oxide during the coloration process, electron injection occurs accompanied by a significant decrease in the valence state of W to maintain the electric neutrality. Therefore, the Al/W molar ratio ($x$) of different colored SERS substrates can be calculated from the quantity of injected charge with the following equation[21]:

$$x = QM/(A\delta e\rho N_A), \qquad (1)$$

where $Q$ denotes the value of inserted charge by Al intercalation at a given potential, $A$ is the electrode surface area, $\delta$ is the film thickness, $M$ is the molar mass, $\rho$ is the density, $e$ is the

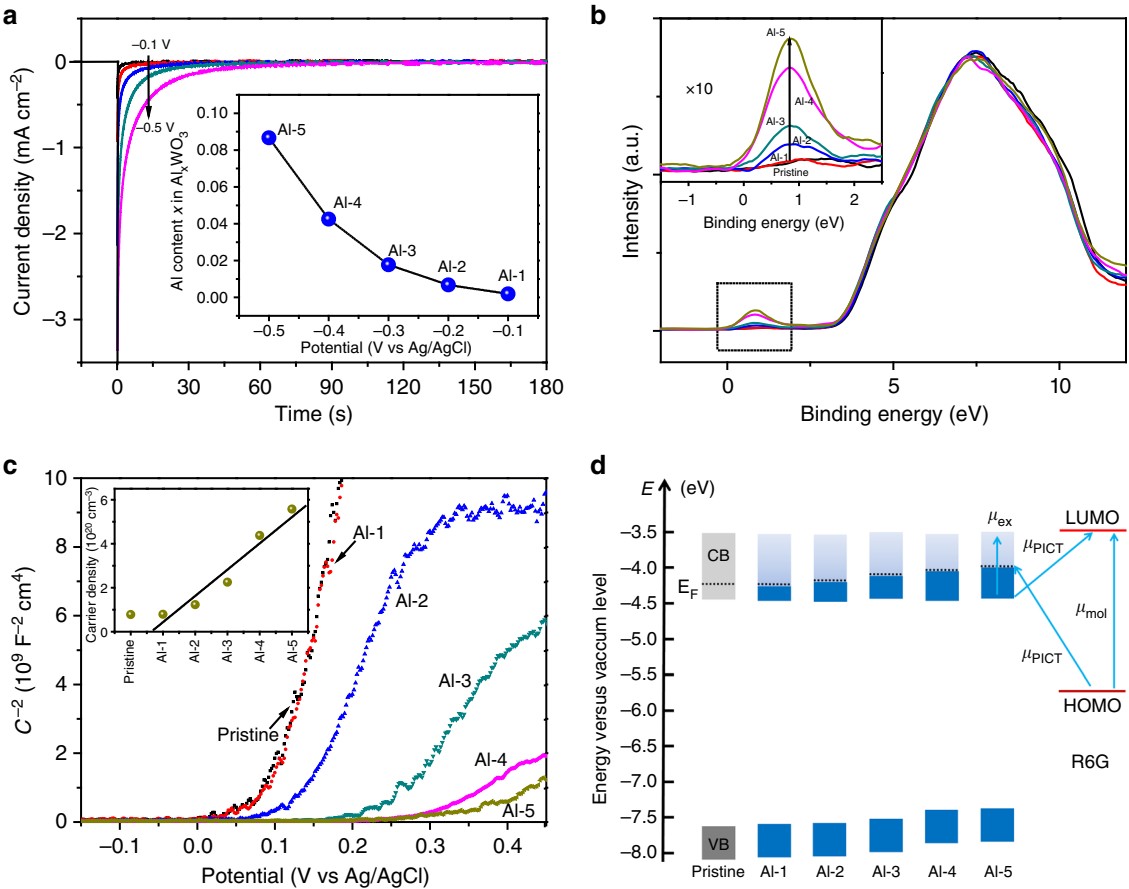

**Fig. 4** Systematic manipulation of the electronic structures. **a** Chronoamperogram of Al-intercalated tungsten oxide films under an applied potential from −0.1 to −0.5 V vs. Ag/AgCl, denoted as Al-1, Al-2, Al-3, Al-4, and Al-5, respectively. Inset shows the number of intercalated Al ions in the $Al_xWO_3$ compositions as a function of applied potential. **b** X-ray photoelectron spectroscopy (XPS) valence-band spectra, and **c** Mott–Schottky plots for the pristine and Al-intercalated tungsten oxide films obtained at different voltages, ranging from −0.1 to −0.5 V. Inset of **b** shows the peaks near the Fermi levels with an expanded Y-axis. Inset of **c** shows density of charge carriers calculated from the Mott–Schottky plots for pristine $WO_3$ and Al-intercalated films. **d** Schematic of band-level diagram for pristine and colored films, illustrating improved charge-transfer transitions between the substrates and analyte molecules (R6G) due to gradual filling of the conduction band (CB) states

elementary charge, and $N_A$ is Avogadro's constant. The calculation shows that there is a monotonic increase in the Al/W molar ratio (from 0.188% to 8.67%) as the Al-intercalation potential is reduced from −0.1 to −0.5 V (Fig. 4a), suggesting that such an electrochromic process does allow control over the chemical composition of the colored SERS substrates, and thus offers a way to control their electronic structures. The adjustability or controllability of electronic structures induced by the electrochromic process can be corroborated by the combination of UV–Vis spectroscopy, XPS, and Mott–Schottky analysis. First, the structure of the valence band near the Fermi level can be determined by XPS analysis. As shown in Fig. 4b, the XPS valence spectra of the colored SERS substrates show a two-region structure: a broad peak situated at a binding energy of about 7.39 eV due to the O 2p state, and a small peak centered at 0.83 eV below the Fermi level. This latter peak corresponds to the lower part of the conduction band associated with the W 5d state, according to the ligand field theory[22]. Notably, for all of the colored and uncolored substrates, the leading edges of the O 2p peaks toward the Fermi level almost completely overlap, indicating that the valence-band character is almost unaffected by coloration/bleaching. In contrast, the intensity of the small peak gradually increases with an increase in the Al/W molar ratio among the samples, suggesting the gradual occupation of the W 5d states by electrons. Second, electrochemical Mott–Schottky

analysis gives more quantitative information about the variations in electronic structure[23–25]. The Mott–Schottky plots of the colored and uncolored substrates at a frequency of 1 kHz in 0.1 M of tetrabutylammonium perchlorate (TBAP) in propylene carbonate are presented in Fig. 4c, where the reciprocal of the square of capacitance ($C^{-2}$) is plotted against the potential (V). Clearly, all of the substrates exhibit n-type characteristics with positive slopes, implying that electrons serve as the main carriers in these substrates (Fig. 4c). From the intercepts of the Mott–Schottky plots, two important electronic parameters, the donor (electron) density ($N_d$) and flat-band potential ($V_{fb}$), of the colored and uncolored substrates can be quantitatively determined using the following relation[24]:

$$\frac{1}{C^2} = \frac{2}{e_0 \varepsilon \varepsilon_0 N_d}[(V - V_{fb}) - kT/e_0],    (2)$$

where $C$ is the specific capacitance (F cm$^{-2}$), $e_0$ is the electron charge ($1.60 \times 10^{-19}$ C), $\varepsilon$ is the dielectric constant of tungsten oxide ($\varepsilon = 20$), $\varepsilon_0$ denotes the permittivity of a vacuum ($8.85 \times 10^{-14}$ F/cm), $V$ is the applied potential (V), $k$ is the Boltzmann constant ($1.38 \times 10^{-23}$ J/K), and $T$ is the absolute temperature (K). Accordingly, the electron density ($N_d$) for the pristine uncolored tungsten oxide, calculated from the slope of the obtained Mott–Schottky plots, is estimated to be $0.78 \times$

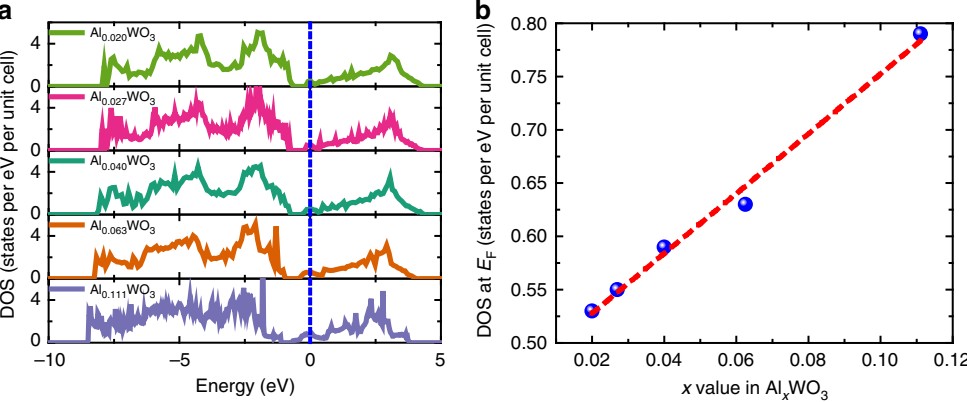

**Fig. 5** Density of states for Al-intercalated tungsten oxide. **a** Calculated density of states (DOS) curves for cubic tungsten bronzes, $Al_xWO_3$, near the Fermi level. The Fermi level is set as 0 eV. **b** Density of states at the Fermi level, for $Al_xWO_3$ with varied $x$

$10^{20}$ cm$^{-3}$, in good agreement with previous studies[24,25], while the $N_d$ values of the colored samples are found to be up to seven times larger than that of the uncolored tungsten oxide, ranging from $0.79 \times 10^{20}$ to $5.43 \times 10^{20}$ cm$^{-3}$. Notably, a roughly linear relation between the electron density and intercalation potential (or Al/W molar ratio) can be established within a certain range (Inset of Fig. 4c), further corroborating the quantitative electron filling in the colored samples observed from the XPS VB spectra (Fig. 4b). At the same time, a gradual shift in the flat-band potential ($V_{fb}$), determined from the Mott–Schottky plots by the intercept on the potential axis, can also be observed in the colored samples, with the Al-5 sample showing the maximum shift of 0.26 eV when compared with pristine uncolored tungsten oxide, with a $V_{fb}$ value of 0.09 eV (vs. Ag/AgCl electrode) (Supplementary Fig. 18). The shift in flat-band potential, which can be used to experimentally determine the Fermi level location[26], further indicates the adjustability of the electronic structures. Finally, by acquiring optical bandgap ($E_g$) values from UV–Vis spectroscopy (Supplementary Fig. 19), the energy-level diagrams can be constructed, indicating that the conduction band minimum (CBM) is roughly constant as the Fermi level is gradually shifted upward, while the valence-band maximum (VBM) linearly related to the extent of Al intercalation (Fig. 4d, Supplementary Table 1). Because the SERS signal is maximized where the derivative of the density of states (DOS) is largest, the charge-transfer transitions in a semiconductor-molecular system are usually considered to start or terminate at band-edge-derived states[7]. Considering that the frequencies of both the interband excitonic transition and the interfacial photo-induced charge transfer (PICT) transition (from VBM to LUMO) in our system are far away from resonance with the incident laser (2.33 eV), they probably contribute little to the overall SERS due to lack of effective band-level alignment[7]. Instead, the quantitatively accumulated electrons at the bottom of the CB probably participate in the PICT process in such metallic semiconductors[27], thus contributing to the quantitative increase in SERS enhancement upon Al intercalation.

The gradually variation in the electronic structure of Al-intercalated tungsten oxide is further verified by the results of density functional theory calculations. For simplicity, these calculations examine a cubic bronze system with Al atoms inserted into the unit cell (Supplementary Methods). As shown by the calculated DOS in Fig. 5a, the Fermi level is located well into the conduction band for the $Al_xWO_3$ ($0 < x < 0.12$) bronzes, rendering them metallic, as noted in other studies[28], in contrast to the electronic behavior of typical semiconductors. With increasing $x$ in $Al_xWO_3$, that is, the insertion of more Al atoms into the

unit cell, the shape of the band structure does not change significantly. Instead, a gradual down-shifting of both CB and VB can be observed with respect to the Fermi level, which is in exact agreement with the experimental band-level diagram shown in Fig. 4d. In particular, a linear relationship can be drawn when plotting the location of CBM against the Al accommodation number $x$ in $Al_xWO_3$, indicating a rationally adjustable energy-band location (Supplementary Fig. 20). Further, an obviously linear increase of the DOS at the Fermi level can also be observed with increasing $x$, as shown in Fig. 5b. This quantitatively modulated DOS near the Fermi level in the colored samples, in great contrast to the usually fixed DOS values shown by common metallic or semiconducting SERS substrates[8,29], is presumably responsible for the unique color-dependent enhancement in SERS. Finally, to illustrate the dependence of signal intensity on DOS for the Raman vibrational bands, Herzberg–Teller theory is applied to the charge-transfer contributions to SERS, derived from Albrecht[30] and Lombardi et al.'s[31] work on a metal-molecular system. As illustrated in detail in Supplementary Methods, the SERS intensity ($I$) is in proportion to the square of the polarizability tensor ($\alpha$), making $I$ directly dependent on the DOS ($\rho$) near the Fermi level of the substrate. The simplest mathematical expression of this relation is as follows:

$$I \propto [\kappa\rho]^2, \tag{3}$$

where $\kappa$ represents the contributions from charge transfer, determined by the band-level alignment between the semiconductor and molecules, and $\rho$ is the DOS at the Fermi level of the semiconductor. Therefore, it can be inferred that the linearly increased DOS in the CB of $Al_xWO_3$ upon charge intercalation is the origin of the quantitative control over its SERS enhancement.

Finally, we try to realize a similar systematic modulation of the electronic structures of tungsten oxide films through pathways other than electrochromic processes, for example, by creating oxygen vacancies. According to our previous work[8], the formation of oxygen defects in thermally reduced samples can in principle be controlled by varying the treatment time (from 20 to 100 min) under a precisely controlled $H_2$ flow at 350 °C, with the aim of gradually modulating the electronic structure. Unfortunately, the series of $H_2$-treated samples shows no systematic trends in the modulation of the bandgap, carrier density, or the filling of W $5d$ states, as determined from the UV–Vis spectra, Mott–Schottky plots, and XPS valence-band spectra (Supplementary Fig. 21). In fact, reproducible control over the electronic properties is barely possible even among batches of $H_2$-treated substrates fabricated under the same

conditions (Supplementary Fig. 22), which probably explains the relatively poor reproducibility among the hydrogen-treated samples as SERS substrates (Supplementary Fig. 14). Further, the morphologically uniform sputtered films as SERS substrates, in contrast to the common drop-casted counterpart (Supplementary Fig. 23, Note 2), are also benefiting from the fluorescence quenching effect (Supplementary Fig. 24, Note 3) and the colorimetric functionality (Supplementary Note 4).

## Discussion

In summary, by means of electrochemical intercalation of cation species ($H^+$, $Li^+$, $Na^+$, $K^+$, $Mg^{2+}$, and $Al^{3+}$), SERS substrates with boosted SERS activity can be obtained through the coloration of electrochromic semiconductors (such as $WO_3$ and NiO), showing high reproducibility, renewability, and unique colorimetric functionality. Upon quantitatively controlled ion intercalation, the electronic structure of the host semiconductor (such as $WO_3$) can be rationally manipulated, with a gradually increasing DOS in the CB, which is quantitatively related to the SERS enhancement, leading to the reproducible and renewable control over the SERS enhancement of the substrates.

## Methods

**Film fabrication**. To fabricate electrochromic film substrates for SERS tests, slabs of FTO-coated glass with planar sizes of $1.5\ cm \times 2.5\ cm$ were used as substrates, which were cleaned beforehand in acetone, ethanol, and distilled water successively under sonication for 20 min before drying at 60 °C. The tungsten oxide thin films on FTO glass were prepared by reactive DC magnetron sputtering at room temperature. The background pressure of the sputtering cavity was evacuated to $8 \times 10^{-4}$ Pa before deposition. Then, the DC power of the tungsten target (>99.9%) was kept at 200 W for 10 min, with the ratio of $Ar:O_2$ fixed as 80:20 sccm and the pressure maintained at 0.2 Pa during sputtering.

**Electrochemical and electrochromic measurements**. The electrochemical behavior was studied using a CHI 660C electrochemical workstation (CH Instruments, Inc., China) at room temperature. A three-electrode quartz cell setup was used, with the films sputtered on FTO glass, Pt wire, and Ag/AgCl/KCl as the working electrodes, counter electrode, and reference electrode, respectively. All potentials reported herein have been measured vs. Ag/AgCl. Aqueous electrolyte solutions of various chlorides (HCl, LiCl, NaCl, KCl, $MgCl_2$, and $AlCl_3$) with a concentration of 1 M have been used for the electrochromic treatments. The pH values of the aqueous solutions containing HCl, LiCl, NaCl, KCl, $MgCl_2$, and $AlCl_3$ are measured to be 0.01, 6.03, 9.15, 4.10, 5.12, and 2.96, respectively. The electrochromic characteristics were examined by a combination of the electrochemical workstation and a UV–Vis optical spectrometer, with alternating biases switching between −0.5 and +0.2 V relative to Ag/AgCl applied to the working electrode, for coloring and bleaching of the film. CV tests were obtained from the film electrode with a scan rate of $10\ mV\ s^{-1}$ in the potential range of −0.6 to +0.4 V in 1 M $AlCl_3$ aqueous analyte. The Mott–Schottky plots were collected from −0.6 to 0.6 V vs. Ag/AgCl at the frequency of 1000 Hz in 0.1 M of TBAP in propylene carbonate saturated with Ar atmosphere. Colored films as substrates were obtained after discharging potentiostatically for 180 s at certain potentials in 1 M aqueous $AlCl_3$ solution.

**Raman measurement**. Typically, SERS spectra of R6G molecules deposited on the pristine and colored films as substrates were obtained under laser excitation at 532.8 nm, after the films were thoroughly rinsed with distilled water, dried at 60 °C and re-loaded with R6G. Specifically, R6G ethanol solutions with concentration varying from $10^{-4}$ to $10^{-6}$ M were obtained from a stock solution of $10^{-3}$ M by successive dilution. Then, 50 μL of R6G solution with a given concentration was dropped on the film, and was allowed to spread over the whole surface of the film spontaneously, followed by drying at room temperature for 2 h in the dark. Raman spectra were subsequently acquired on a high-resolution confocal Raman spectrometer (LabRAM HR-800) with an excitation laser of 532 nm. The spectra were collected by using a ×50 L objective lens for 15 s with a laser spot diameter of about 1 μm and power of 0.3 mW for all acquisitions. Raman spectra from different locations were collected for each sample, with the signal intensity averaged for final analysis to estimate the RSD values for the EFs.

**Reporting summary**. Further information on experimental design is available in the Nature Research Reporting Summary linked to this article.

## Data availability

The data that support the findings of this study are available from the corresponding authors on reasonable request. Additionally, data reported herein have been deposited in the Figshare database, and are accessible through https://doi.org/10.6084/m9.figshare.7577444.

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

## Acknowledgements

This work was supported by the National Natural Science Foundation of China (51772319, 51772320, 51572286, and 21503266) and the Outstanding Youth Fund of Jiangsu Province (BK20160011), the Youth Innovation Promotion Association, CAS (2018356) the Natural Science Foundation of Jiangxi Province (20181ACB20011), and the Science and Technology Project of Nanchang (2017-SJSYS-008).

## Author contributions

Z.Z. conceived the project. S.C. designed the experiments and analyzed the data. S.C. and Z.W. performed material synthesis, structural characterization, and Raman measurements. W.G. contributed with the parts of simulation and calculation. S.C. and Z.Z. co-wrote the paper. Z.C., W.L., and J.R.L. contributed in discussions and comments on the manuscript.

## Additional information

**Competing interests:** The authors declare no competing interests.

