## [Peer Review File · Nature Communications]

Reviewers' Comments:

Reviewer #1:

Remarks to the Author:

I recommend publishing the study entitled "Electrochromic Semiconductor SERS Substrates with High Colorimetric Functionality, Reproducibility, and Renewability" after minor revision. It describes novel substrates for SERS with a lot of necessary details. Nevertheless, several issues should be solved; some minor corrections are recommended.

Several suggestions are given below.

- 1) Firstly, the key issue to be solved is the lack of information about the specific molecule used, i.e. Rhodamine 6G (R6G), which is mentioned firstly on the row 231 in the part Results. It is not introduced in the abstract and in the introduction. I guess, the distinctive physical (mainly optical) and chemical properties (pi electronic skeleton and substituents important for chemisorption) are very important for the obtained results. The absorption maximum of R6G (at ca. 530 nm – see spectral data <https://omlc.org/spectra/PhotochemCAD/html/083.html> or ca. 535 nm – see https://pubchem.ncbi.nlm.nih.gov/compound/rhodamine_6g#section=Chemical-and-Physical-Properties) fits the only one used excitation wavelength (532.8 nm) of Raman scattering measurements. In fact, it fulfills the condition to measure resonance Raman spectra in solution or surface-enhanced resonance Raman spectra (SERRS) on a SERS-active substrate. The interaction of the very specific energy states of R6G with the incident photons and states of electrochromic semiconductor substrates should be very important and these aspects should be emphasized in the manuscript. This quite simple modification of the text should be acceptable. Nevertheless, for general use of the novel substrates, it will be important to measure Raman scattering spectra of R6G outside the molecular resonance (or even better to measure the excitation profiles of SERS/SERRS) and/or to study another (colorless or differently colored) molecule. Such measurements enable to distinguish the role of the electrochromic substrate, the molecular optical properties and their mutual interactions. R6G is highly fluorescent species, the adsorption on the substrate causes probably very important fluorescence quenching. (Rhodamine 6G is used as a laser dye in dye lasers, and is pumped by the second harmonic from an Nd:YAG laser (ca. 532 nm), nitrogen laser, or argon ion laser. The dye has a remarkably high photostability, high fluorescence quantum yield (ca. 0.95), low cost, and its lasing range has close proximity to its absorption maximum (approximately 530 nm). The lasing range of the dye is 570 to 660 nm with a maximum at 590 nm.) The optical properties of R6G (both in solid state and in solution) complicates the reliable calculation of the enhancement factor.
- 2) Commercial Klarite substrates are in fact quite poor SERS substrates. There are many other commercial substrates (for examples see a recent review <https://www.ncbi.nlm.nih.gov/pmc/articles/PMC5485789/>). I guess other plasmonic substrates can be compared with novel semiconductor samples. I understand that the drops of analyte solution can be placed on Klarite substrates as well as on semiconductor samples prepared. However, in such cases, both the concentration, the volume of the droplet and the examined area should be discussed together. The concentration LOD value is misleading because the measurement is not performed in (colloidal) solution. The amount of examined molecule could be changed by the selection of the droplet volume without the change of solution concentration. Please, modify the text from the point of view of both different plasmonic substrates and the use of droplet deposition.
- 3) The removal of R6G should be described clearly discussing the renewability. The surface purification is unclear. Of course, it will be nice to see alternate measurements of two different analyte to display the quality of the purification step and the repeatability of SERS spectra of both analytes. Nevertheless, at least the purity of the surface after R6G (prior the next use of the substrate) has to be demonstrated.
- 4) The colorimetric detection of SERS activity mentioned in the manuscript is questionable. It should be emphasized the relationship between colorimetric detection and the excitation wavelength used for

- SERS experiments. For example, see the sentence "Impressively, upon charging, the deep-blue-colored substrate shows a striking increase of SERS enhancement compared with the uncolored substrate" – the color of the substrate should be related to the excitation wavelength used.
- 5) The abbreviations (e.g. ITO, FTO) should be explained at their first usage in the manuscript.
 - 6) Row 142 – Aqueous solution should be described more precisely – pH?, the concentration of cations?, the concentration of anions? (which type?)
 - 7) Rows 150/181 – What is a flat surface (what is the range of roughness?)? The range of roughness should be related to the thickness of the sputtered film (row 153). Based on supplementary data, it looks like that the variability is 80 nm (plus/minus 40 nm) which is quite a lot considering 300 nm of film thickness.
 - 8) Rows 159 – 164 What is characterization and/or identification? The characteristics of both amorphous and crystalline tungsten oxide films should be described clearly with references considering the difference of thin film and bulk materials.
 - 9) Row 193 What is "good reversibility"?
 - 10) The elemental mapping in all figures (including supplementary) is hardly visible? Please, change the settings of contrast and brightness.
 - 11) Row 248 I guess the "plateau" is questionable.
 - 12) The electrochromic coloration should be related to excitation profiles. I do not require excitation profiles to be supplemented in the manuscript, but their role should be noted.
 - 13) Row 258 The LOD calculation in the form of concentration is not fully suitable for the type of substrates used (see above).
 - 14) Row 321/322 The same value of absorbance at 532 nm is not a confirmation of the same coloration. The shape of the band should be important.
 - 15) The cycles of coloring/decoupling and photocatalytic treatment should be described more carefully especially from the point of view of the (repeatable) deposition of R6G. The physisorption or chemisorption of R6G should be specified.
 - 16) I am not sure, if the term "regeneration" is correct. It looks like cycles, but what is regenerated. How the surface is cleaned and how the cleaning quality is evaluated?
 - 17) Row 444 The DOS abbreviation should be explained in the text (not only in the figure caption).
 - 18) Row 485 The term "colorimetric enhancement" is rather misleading; "colorimetry" is an analytical technique.
 - 19) Rows 553 – 555 What is the analyte? See the sentence "Colored films as SERS substrates were obtained after discharging potentiostatically for 180 s at certain potentials in an aqueous analyte of 1 M of chloride salt. Is it in the presence of R6G in the form of chloride? It looks peculiarly. Please, specify the chloride salt.
 - 20) The Figure S11 is described as "The absorbance of the pristine and colored tungsten oxide films monitored at 532 nm." However, the horizontal axis of the graph is wavelength, i.e. it is not a monitoring of absorbance at one value of wavelength.
 - 21) The Figures S12 b is denoted as "(b) EF values calculated from the intensities ...", the vertical axis in the figure is described as Intensity.
 - 22) The Figure S15 The axis description is "Substrate number". Please, correct typing.
 - 23) Bulk R6G crystals on bare Si/SiO₂ wafer should be described more precisely considering their use for EF calculations.

Reviewer #2:

Remarks to the Author:

The authors introduce electrochromic WO₃ substrates as efficient and renewable SERS substrates. Here the advantage of the electrochromic behaviour is that the change in absorbance can be related to the SERS enhancement, and this can be used for enhancement calibration.

I have some general, and also few detailed questions:

- How is the SERS enhancement factor calculated in detail? The authors should show the reference spectrum, and give the values for the calculation.
- The elemental maps in my documents are essentially black, I cannot discern anything there.
- How does the regeneration work exactly? Are the analyse molecules etched away during the electrochemical process? To prove this, the authors should show the spectra before and after regeneration, which means with the molecules at a given concentration, and then after the electrochromic regeneration of the "clean" , but already intercalated sample. This would show that no trace amounts of the previous analyse remain on the substrate.
- It would be nice to see that the SERS enhancement works also with other molecules than RG6.
- What are the restrictions of SERS with semiconductor substrates? Does the excitation have to be above the band gap of the molecule? What are the resonance conditions? Would it also work at 785 nm?
- How are the samples treated after the electrochemical intercalation? Are they washed somehow, or just left to dry? This should be described in detail. In Figure 1c is the absorbance recorded in solution? And it would be nicer to have color photos in the insets of 1c.
- How can the effect that more SERS effective substrate can be distinguished from less effective one by the naked eye be useful? After all, it is the Raman spectrum from the molecule that is of interest, and the substrate color is not sensitive to the signal from the molecule.

Detailed comments:

- Does Figure 1b show 50 curves that fall exactly onto each other? The caption suggests that.
- An AFM image at the same magnification as the SEM image in Fig 1a should be supplied to get insight into the grains of the film.
- In Figure 2b I find it daring to speak of a plateau at -1.2. Then, why do the lower 3 data points do not have error bars? What is the goodness of a linear fit to that data set?
- The comparative figures 3b and S13 should have values at the y axis, such that absolute signal versus deviation becomes evident.
- It would be better to give the absorbance in meaningful units such as optical density and not in a.u.
- substrate description jumps between ITO and FTO
- On page 13 they say "only few studies have reported truly renewable SERS substrates", these should be cited here.
- pages should be numbered.

The paper can be written in a more concise fashion, introduction is a bit repetitive, with I do not know how many times "uniformity, reproducibility, and renewability".

Also, when the sample description starts, first 6 supporting Figure are discussed before the first main text figure. For the reader it would be nicer to come more straight to the point.

To my opinion, the manuscript needs a major revision with substantial additional experimental data before it can be reconsidered for publication in Nat. Commun.

Our response to the editorial requests:

We would like to thank the editor and reviewers for their deep and thorough reviewing of our manuscript. In view of these valuable queries, we have further revised our manuscript, with the changes marked out in the main text and also listed as follows. Here are the detailed responses to the comments.

REQUESTS BY REVIEWERS:

Reviewer 1:

I recommend publishing the study entitled “Electrochromic Semiconductor SERS Substrates with High Colorimetric Functionality, Reproducibility, and Renewability” after minor revision. It describes novel substrates for SERS with a lot of necessary details. Nevertheless, several issues should be solved; some minor corrections are recommended.

Several suggestions are given below.

Author reply: We are grateful to the “reviewer 1” for the positive and encouraging comments. Also, the issues raised by the reviewer have been meticulously examined and solved one by one as detailed below.

Comment 1:

1) Firstly, the key issue to be solved is the lack of information about the specific molecule used, i.e. Rhodamine 6G (R6G), which is mentioned firstly on the row 231 in the part Results. It is not introduced in the abstract and in the introduction. I guess, the distinctive physical (mainly optical) and chemical properties (pi electronic skeleton and substituents important for chemisorption) are very important for the obtained results. The absorption maximum of R6G (at ca. 530 nm – see spectral data <https://omlc.org/spectra/PhotochemCAD/html/083.html> or ca. 535 nm – see https://pubchem.ncbi.nlm.nih.gov/compound/rhodamine_6g#section=Chemical-and-Physical-Properties) fits the only one used excitation wavelength (532.8 nm) of Raman scattering measurements. In fact, it fulfills the condition to measure resonance Raman spectra in solution or surface-enhanced resonance Raman spectra (SERRS) on a SERS-active substrate. The interaction of the very specific energy states of R6G with the incident photons and states

of electrochromic semiconductor substrates should be very important and these aspects should be emphasized in the manuscript. This quite simple modification of the text should be acceptable. Nevertheless, for general use of the novel substrates, it will be important to measure Raman scattering spectra of R6G outside the molecular resonance (or even better to measure the excitation profiles of SERS/SERRS) and/or to study another (colorless or differently colored) molecule. Such measurements enable to distinguish the role of the electrochromic substrate, the molecular optical properties and their mutual interactions. R6G is highly fluorescent species, the adsorption on the substrate causes probably very important fluorescence quenching. (Rhodamine 6G is used as a laser dye in dye lasers, and is pumped by the second harmonic from an Nd:YAG laser (ca. 532 nm), nitrogen laser, or argon ion laser. The dye has a remarkably high photostability, high fluorescence quantum yield (ca. 0.95), low cost, and its lasing range has close proximity to its absorption maximum (approximately 530 nm). The lasing range of the dye is 570 to 660 nm with a maximum at 590 nm.) The optical properties of R6G (both in solid state and in solution) complicate the reliable calculation of the enhancement factor.

Author reply: We greatly appreciate the valuable suggestions.

(1) About the information of Rhodamine 6G (R6G). The information about R6G has been introduced in the abstract and in the introduction. Especially, further details on the distinctive physical (mainly optical) and chemical properties (π electronic skeleton and substituents important for chemisorption) of R6G, including its absorption and fluorescence emission spectra and molecular structure, are discussed in the Supporting Information (Supplementary Note S1), as also described below.

“Rhodamine 6G (R6G) is a well-known laser dye, featured by its high photostability, high fluorescence quantum yield (ca. 0.95) and low cost. Structurally, R6G belongs to the family of xanthenes, which is featured by the bulky monoethylamino group with methyl as ortho-substituents in the xanthene skeleton, together with carboxylate ester (COOEt) group in the lateral phenyl ring (Figure S8a). The carboxylate ester groups contribute toward the mobility of π -electron of the xanthene skeleton, resulting in the resonance structures of the xanthene moieties upon excitation. With the contributions from such resonance structures, the positive charge in the cationic R6G dye molecule can be

stabilized across the 9-carbon atom in xanthene skeleton, showing good affinity to the negatively charged tungstic units existing in electrochromic SERS substrates. Also, it is noted that the maximum absorption for R6G in aqueous solution is observed at ca. 530 nm (Figure S8b), which exactly fits the used excitation wavelength (532.8 nm) during SERS measurements. Accordingly, when a R6G system is excited by 532 nm pulsed laser, it shows a molecular resonance Raman effect in addition to the normal SERS effect, which leads to larger SERS enhancements. Thus, the so-called surface-enhanced resonance Raman scattering (SERRS) effect could be recognized in our system, basing on the coincidence of incident photon energy and electronic transition in highly fluorescent R6G molecules.”

Figure S8. (a) Molecular structure and (b) absorption spectrum of R6G.

(2) About the SERRS effect. As suggested by the reviewer, the molecular transition between the HOMO and LUMO levels of R6G at 2.3 eV is energetically near that of the excitation laser (532 nm), which provides an important resonant pathway for the observed SERS enhancement. In fact, such thermodynamically feasible resonance can contribute to the overall Raman enhancement through intensity borrowing based on the Herzberg-Teller vibronic coupling, according to the theory established by Lombardi et al (Chem. Rev., 2016, 116(24): 14921-14981). On coupling, the relatively weak charge-transfer resonance borrows intensity from the stronger nearby resonances, such as molecular and exciton transitions, as can be expressed by Herzberg-Teller coupling terms. In this case, the so-called surface-enhanced resonance Raman scattering (SERRS) effect could be recognized, basing on the coincidence of incident photon energy and electronic transition

in R6G molecules. The important interaction of the very specific energy states of R6G with the incident photons and states of electrochromic semiconductor substrates has been emphasized in the revised manuscript.

“Further investigation demonstrates that the electrochromic SERS substrates are also effective for other types of analytes such as crystal violet (CV) and Victoria blue B (VBB) with 532 nm laser excitation (Supplementary Fig. 11). Furthermore, in the case that the laser excitation is not resonant with the molecular absorbance of R6G, the SERS signals from R6G are still distinguishable under 633 nm laser excitation (Supplementary Fig. 12), suggesting the generality of this application of the electrochromic SERS substrates. It should be noted that the Raman signals for CV and VBB are relatively weaker as the excitation wavelength (532 nm) is not in direct resonance with the absorption of CV at about 590 nm or VBB around 599 nm. In contrast, the molecular transition between the HOMO and LUMO levels of R6G at 2.3 eV is energetically near that of the excitation laser (532 nm), which provides an important resonant pathway for the observed SERS enhancement. In fact, such thermodynamically feasible resonance can contribute to the overall Raman enhancement through intensity borrowing based on the Herzberg-Teller vibronic coupling, according to the theory established by Lombardi et al (Chem. Rev., 2016, 116(24): 14921-14981). On coupling, the relatively weak charge-transfer resonance borrows intensity from the stronger nearby resonances, such as molecular and exciton transitions, as can be expressed by Herzberg-Teller coupling terms. Accordingly, the so-called surface-enhanced resonance Raman scattering (SERRS) effect could be recognized in the R6G/tungsten oxide system, basing on the coincidence of incident photon energy and electronic transition in R6G molecules.”

Figure S11. SERS spectra of (a) crystal violet (10^{-3} M) and (b) Victoria blue B (10^{-3} M) on the pristine (bleached) and Li-intercalated tungsten (colored) oxide films under 532 nm laser excitation. A comparison of SERS activity between the pristine (bleached) and Li-intercalated (colored) tungsten oxide film under 532 nm laser excitation, for (c) vibrational band at 913 cm^{-1} of CV and (d) vibrational band at 1617 cm^{-1} of VBB. Cation intercalation is conducted in 1 M aqueous LiCl analyte solution via chronoamperometry at constant potential of -0.5 V for 180 s.

Figure S12. (a) SERS spectra of R6G (10^{-4} M) on the pristine (bleached) and Li-intercalated tungsten oxide films (colored) under 633 nm laser excitation. A comparison of SERS activity between the pristine (bleached) and Li-intercalated (colored) tungsten oxide film under 633 nm laser excitation for vibrational band at 612 cm^{-1} of R6G. Cation intercalation is conducted in 1 M aqueous LiCl analyte solution via chronoamperometry at constant potential of -0.5 V for 180 s.

(3) About other analytes or excitation wavelengths. To further distinguish the role of the electrochromic substrates, the SERS spectra acquired from other types of differently colored analytes, such as crystal violet (CV) and victoria blue B (VBB) on the electrochromic SERS substrates under 532 nm laser excitation are presented in Supplementary Fig. 11 (the physicochemical properties of the analytes are detailedly outlined in Supplementary Note S1). Moreover, the SERS signals from R6G are also measured at other laser excitation wavelengths (such as 633 nm) (Supplementary Fig. 12). The results indicate that SERS signals of CV and VBB adsorbed on the electrochromic SERS substrates can also be detected under 532 nm laser excitation, but are relatively weaker as the excitation wavelength (532 nm) is not in direct resonance with the absorption of CV at about 590 nm or VBB around 599 nm. As mentioned before, an explanation for this phenomenon has been offered in the revised manuscript.

(4) About the fluorescence quenching. As suggested by the reviewer, the importance of fluorescence quenching based on our electrochromic substrates has been stressed in the revised manuscript (Supplementary Note S2).

“Rhodamine 6G (R6G) is known to be a highly fluorescent dye, featured by its high photostability, high fluorescence quantum yield (ca. 0.95) and low cost. It is observed that pure R6G dye exhibits a strong fluorescent emission at 551 nm in solution (Figure S24a), which would prevent observation of the SERS spectrum. However, after the loading of R6G dye on the electrochromic substrate, the drastic quenching in the fluorescence intensity of R6G occurs with respect to the reference spectrum on SiO_2/Si substrate (Figure S24b). The efficient fluorescence quenching for adsorbed analyte is probably ascribed to the metallic nature of the electrochromic substrates with high electron

conductivity, which will facilitate the electron transfer or energy transfer to alleviate the charge-carrier recombination. (Phys. Rev. Lett. 2002, 89, 203002)

Figure S24. (a) Fluorescence spectrum of R6G with the excitation wavelength of 532 nm. (b) Raman spectra of R6G (10^{-4} M) on the electrochromic tungsten oxide film and bare SiO₂/Si substrate under 532 nm laser excitation, without baseline correction. The electrochromic tungsten oxide film is prepared in 1 M aqueous AlCl₃ analyte solution via chronoamperometry at constant potential of -0.5 V for 180 s.

(5) About the calculation of the enhancement factor. The calculation of the enhancement factor (EF) in our work is detailed as follows:

$$EF = (I_{\text{SERS}}/N_{\text{SERS}})/(I_{\text{bulk}}/N_{\text{bulk}}) \quad (1)$$

$$N_{\text{SERS}} = CVN_{\text{A}}A_{\text{Raman}}/A_{\text{Sub}} \quad (2)$$

$$N_{\text{bulk}} = \rho h A_{\text{Raman}} N_{\text{A}}/M \quad (3)$$

where I_{SERS} and I_{bulk} are the intensities of the selected Raman peak in the SERS and normal Raman spectra, and N_{SERS} and N_{bulk} are the average number of molecules in the scattering area of the SERS and non-SERS substrates, respectively. Here the data for R6G (0.05 M) on bare Si/SiO₂ substrate were used as non-SERS-active reference. Specifically, the intensity was obtained by taking average from measurements of 20 spots, and the number of analyte molecules (N_{SERS}) was estimated by Supplementary equation 2 on the assumption that the analyte molecules were distributed uniformly on the substrates. C is the molar concentration of the analyte solution, V is the volume of the droplet, and N_{A} is the Avogadro constant. A_{Raman} is the laser spot area (1 μm in diameter) of the Raman

scanning. 20 μL of the analyte solution was spontaneously spread into a circle of about 3 mm in diameter on the substrate after solvent evaporation, from which the effective area of the substrate, A_{Sub} , can be obtained. On the other side, N_{bulk} can be calculated by Supplementary equation 3 on the basis of molecular weight (M) and density (ρ) of bulk R6G (1.15 g cm^{-3}) and the confocal depth h ($23.64 \mu\text{m}$) of the laser beam (Figure S27).

- (6) Several points concerning the calculation of EF should be emphasized: (1) The fluorescence of R6G dye has been effectively quenched by the electrochromic substrates, which does not prevent observation of the SERS spectrum or the calculation of EF. (2) According to the theory established by Lombardi et al (Chem. Rev., 2016, 116(24): 14921-14981), the molecular resonance of R6G can also contribute to the overall Raman enhancement through intensity borrowing based on the Herzberg-Teller vibronic coupling. On coupling, the relatively weak charge-transfer resonance borrows intensity from the stronger nearby resonances, which can be expressed by Herzberg-Teller coupling terms for intensity borrowing from molecular and exciton transitions, respectively. As a matter of fact, a similar calculation for R6G dye has been made by recent SERS literatures (Nanoscale, 2013, 5(7): 2784-2789.; J Phys. Chem. C, 2012, 116(5): 3320-3328.; Surf. Sci., 1998, 406(1-3): 9-22.; Anal. Chem., 2017, 89(21): 11765-11771.). (3) Although the situation in a real micro-Raman experiment may be complicated, the calculation of enhancement factor is still necessary for the investigation of substrate materials, since it may be the only reference we can appeal for the performance evaluation between different substrates.

Comment 2:

2) Commercial Klarite substrates are in fact quite poor SERS substrates. There are many other commercial substrates (for examples see a recent review <https://www.ncbi.nlm.nih.gov/pmc/articles/PMC5485789/>). I guess other plasmonic substrates can be compared with novel semiconductor samples. I understand that the drops of analyte solution can be placed on Klarite substrates as well as on semiconductor samples prepared. However, in such cases, both the concentration, the volume of the droplet and the examined area should be discussed together. The concentration LOD value is misleading because the

measurement is not performed in (colloidal) solution. The amount of examined molecule could be changed by the selection of the droplet volume without the change of solution concentration. Please, modify the text from the point of view of both different plasmonic substrates and the use of droplet deposition.

Author reply: Thanks for the valuable comment.

(1) About other commercial substrates. We have done our best to find two other commercial substrates, Q-SERS and SERS-SP. The SERS performances of the two commercial substrates are evaluated by the signals of R6G in terms of their SERS activity, reproducibility and reusability. It is found that the batch-to-batch RSDs for Q-SERS and SERS-SP are about 29.6% and 38.4%, respectively, while the one-time use of recycled substrates causes dramatical decrease in signal intensity for both commercial substrates. Clearly, these SERS performances of two commercial substrates are much inferior to that of our electrochromic SERS substrate. The relevant information has been included in the revised manuscript (Figure S25).

Figure S25. Average intensity of the vibrational band (612 cm^{-1}) of R6G (10^{-4} M) under 532 nm laser excitation obtained from two types of commercially available substrates: (a) SERS-SP and (b) Q-SERS substrates. Signal intensities on five SERS substrate in different batches are acquired to calculate the batch-to-batch RSD values, which are 29.6% and 38.4% for Q-SERS and SERS-SP, respectively. It is also found that the one-time use of recycled substrates causes dramatical decrease in signal intensity for both commercial substrates. Note that all samples for SERS measurement are prepared by casting $20\text{ }\mu\text{l}$ of 10^{-4} M R6G in ethanol onto the surface of SERS substrates (area: 4×4

mm²), allowing the solvent to evaporate. To ensure equal amount of analyte loading, the generated droplets have almost the same volume when using the microburets.

(2) About measurement conditions. The experimental details about the sample processing on SERS substrates, including the concentration, the volume of the droplet and the examined area have been detailedly described in the revised manuscript.

“All samples for SERS measurement are prepared by casting 20 μl of 10⁻⁴ M R6G in ethanol onto the surface of SERS substrates (area: 4×4 mm²), allowing the solvent to evaporate. To ensure equal amount of analyte loading, the generated droplets have almost the same volume when using the microburets.”

(3) About the LOD. As suggested by the reviewer, the misleading value of LOD calculated by the IUPCA method has been corrected in the revised manuscript.

Comment 3:

3) The removal of R6G should be described clearly discussing the renewability. The surface purification is unclear. Of course, it will be nice to see alternate measurements of two different analyte to display the quality of the purification step and the repeatability of SERS spectra of both analytes. Nevertheless, at least the purity of the surface after R6G (prior the next use of the substrate) has to be demonstrated.

Author reply:

Thanks for the valuable comment.

In the revised manuscript, each step of the renewability or purification has been carefully monitored by two other means (UV-Vis and Raman spectroscopies) allowing us to validate the electrochromic technique. The relevant results have been included in the revised manuscript, as also described below.

“The efficacy of our electrochromic renewability scheme is also examined by two other means (UV-Vis and Raman spectroscopies) (Supplementary Fig. 17). It should be observed that the spectroscopic profiles obtained from the regenerated SERS substrate after electrochromic coloration are almost identical to the those obtained from the freshly prepared substrate, along with the absence of clearly discerned signals ascribed to R6G analyte,

indicating good renewability that makes the surface able to recover.”

Figure S17. (a) UV-Visible absorption spectra and (b) Raman spectra at 532 nm laser excitation of a series of substrates including the pristine tungsten oxide film, the tungsten oxide film loaded with R6G analytes (10^{-6} M) and the regenerated tungsten oxide films with different regenerative cycles by means of electrochromic treatments.

Comment 4:

4) The colorimetric detection of SERS activity mentioned in the manuscript is questionable. It should be emphasized the relationship between colorimetric detection and the excitation wavelength used for SERS experiments. For example, see the sentence “Impressively, upon charging, the deep-blue-colored substrate shows a striking increase of SERS enhancement compared with the uncolored substrate” – the color of the substrate should be related to the excitation wavelength used.

Author reply: Thanks for the valuable suggestion. The relationship between colorimetric detection and the excitation wavelength used for SERS experiments has been emphasized in the revised manuscript.

The sentence “Impressively, upon charging, the deep-blue-colored substrate shows a striking increase of SERS enhancement compared with the uncolored substrate” has been changed to “Impressively, upon charging, the deep-blue-colored substrate shows a striking increase of SERS enhancement compared with the uncolored substrate towards rhodamine 6G (R6G), crystal violet (CV) and Victoria blue B (VBB), when the excitation wavelength is fixed to be 532 nm (The physicochemical properties of the analytes are detailedly outlined

in Supplementary Note S1)” Similar modification has also been done in other parts of the revised manuscript.

Comment 5:

5) The abbreviations (e.g. ITO, FTO) should be explained at their first usage in the manuscript.

Author reply: Thanks for the valuable suggestion. The abbreviations have been explained at their first usage in the revised manuscript.

Comment 6:

6) Row 142 – Aqueous solution should be described more precisely – pH?, the concentration of cations?, the concentration of anions? (which type?)

Author reply: Thanks for the valuable suggestion. As suggested by the reviewer, a more precise description of aqueous solution has been included in the revised manuscript.

“Aqueous electrolyte solutions of various chlorides (HCl, LiCl, NaCl, KCl, MgCl₂ and AlCl₃) with a concentration of 1 M have been used for the electrochromic treatments. The pH values of the aqueous solutions containing HCl, LiCl, NaCl, KCl, MgCl₂ and AlCl₃ are measured to be 0.01, 6.03, 9.15, 4.10, 5.12 and 2.96, respectively.”

Comment 7:

7) Rows 150/181 – What is a flat surface (what is the range of roughness)? The range of roughness should be related to the thickness of the sputtered film (row 153). Based on supplementary data, it looks like that the variability is 80 nm (plus/minus 40 nm) which is quite a lot considering 300 nm of film thickness.

Author reply: Thanks for the valuable suggestion. We are sorry for the inaccurate description of “a flat surface”, which has been changed in the revised manuscript. It should be stressed that the sputtering deposition can usually give the uniform films tightly sticking to the substrate due to the momenta of the sputtered particles, while the films prepared by dropping or casting the solution to a substrate are rather ununiform and uneven, with large surface roughness and coffee-ring formation. The relevant information has been

included in the revised manuscript (Supplementary Note S3).

Morphologically, the sputtered tungsten oxide film before coloration shows a rather uniform surface, which is constructed by closely-packed coral-like structures in the size range of 200-400 nm (Figure 1a). After coloration, the surface structure of the films remains the same as the pristine film (Figure 1c). In contrast, the tungsten oxide film prepared by drop casting method shows the rough, loose and porous surface feature (Supplementary Figure S26).

Figure S26. (a) Top-view and (b) cross-sectional SEM images of the tungsten oxide film prepared by drop-casting method.

Comment 8:

8) Rows 159 – 164 What is characterization and/or identification? The characteristics of both amorphous and crystalline tungsten oxide films should be described clearly with references considering the difference of thin film and bulk materials.

Author reply: Thanks for the valuable suggestion. The techniques of X-ray diffraction (XRD), Raman spectroscopy, and X-ray photoelectron spectroscopy (XPS) has been employed to investigate the structural characteristics of tungsten oxide films in our work. As an example of crystalline tungsten oxide, the commercially available tungsten oxide material (monoclinic) has also been extensively characterized using the above techniques, and compared with its amorphous counterpart in the revised manuscript, with important references cited (Figure S3).

Figure S3. (a) XRD pattern, (b) Raman spectrum, and (c) XPS spectrum of the commercially available crystalline tungsten oxide powder (monoclinic). The XRD pattern of the commercial tungsten oxide powder corresponds to the monoclinic structure of WO_3 (lattice parameters: $a = 7.297 \text{ \AA}$, $b = 7.539 \text{ \AA}$, $c = 7.688 \text{ \AA}$, $\beta = 90.91^\circ$, JCPDS no: 43-1035), showing typical crystalline characteristics with a series of prominent diffraction peaks at 2θ angles of $20\text{--}80^\circ$. Raman bands occur at 717 (O-W-O vibration) and 807 cm^{-1} (W-O-W stretching), which further confirm the monoclinic structure of the commercial tungsten oxide powder. XPS analysis suggests the coexistence of W in its +5 and +6 oxidation states in the commercial tungsten oxide powder, although the ratio of $\text{W}^{5+}/\text{W}^{6+}$ is quite small ($10.7 : 89.3$).

Comment 9:

9) Row 193 What is “good reversibility”?

Author reply: Thanks for the valuable comment. Detailed description about the “good reversibility” has been included in the revised manuscript.

“It is clearly shown that after 50 cycles of coloring/bleaching operation, the CV curves are substantially unchanged, implying good reversibility between the coloring and bleaching states.”

Comment 10:

10) The elemental mapping in all figures (including supplementary) is hardly visible? Please, change the settings of contrast and brightness.

Author reply: Thanks for the valuable comment. The blurry elemental mapping images have been replaced both in the revised manuscript and supplementary file.

Comment 11:

11) Row 248 I guess the “plateau” is questionable.

Author reply: Thanks for the valuable comment. The improper description “plateau” has been replaced by the word “maximum” in the revised manuscript.

“reaching a maximum within the absorbance range of 0.3-1.2, nearly 28 times greater than that of the pristine uncolored film”

Comment 12:

12) The electrochromic coloration should be related to excitation profiles. I do not require excitation profiles to be supplemented in the manuscript, but their role should be noted.

Author reply: Thanks for the valuable comment.

As suggested by the reviewer, the SERS signals from R6G are also measured at other laser excitation wavelengths (Figure S12). The results indicate that SERS signals of R6G adsorbed on the electrochromic SERS substrates can also be detected under 633 nm laser excitation, but are relatively weaker than at the excitation wavelength of 532 nm. Then, it can be deduced that the prominent enhancement in Raman intensities can probably be achieved at excitation around 532 nm, which may be largely ascribed to the SERRS effect as discussed above. Moreover, the electrochromic coloration in substrate may also contribute to the overall SERS enhancement. As shown in Figure S9, although the peak absorption of the colored substrate varies from the NIR to the visible region when obtained via Al-intercalation at various negative potentials, a roughly linear increase in the absorption monitored at 532 nm can be obtained, which is in a direct relationship with the observed linearly increased EFs as modulated by negative potentials (Figure 2b). The relationship between the SERS enhancement and the optical properties of the colored substrates has been further illustrated by the variation in the DOS (ρ) near the Fermi level of the substrate upon ion-intercalation. The roles of the excitation wavelength and SERRS effect have been emphasized in the revised manuscript.

Comment 13:

13) Row 258 The LOD calculation in the form of concentration is not fully suitable for the type of substrates used (see above).

Author reply: Thanks for the valuable comment. As suggested by reviewer, the LOD estimated using a method recommended by the International Union of Pure and Applied Chemistry (IUPAC) has been removed in the revised manuscript. Instead, the visual evaluation is adopted for the estimation of detection limit for R6G on MOFs, that is, a rough detection limit for R6G is determined from the lowest concentration at which the signals of analytes can still be clearly noticed, which has been commonly used in the recent SERS literatures (Nat. Mater., 2017, 16(9): 918.; Nat. Commun., 2017, 8: 14903.; J. Am. Chem. Soc., 2018, 140(28): 8696–8704.).

Comment 14:

14) Row 321/322 The same value of absorbance at 532 nm is not a confirmation of the same coloration. The shape of the band should be important.

Author reply: Thanks for the valuable comment. The statement has been corrected in the revised manuscript.

“Following this principle, 50 batches of tungsten oxide substrates are colored to the same degree (duplicate spectra at the same intensity and line shape) by means of electrochemical insertion of Al^{3+} ions at -0.5 V.”

Comment 15:

15) The cycles of coloring/decoring and photocatalytic treatment should be described more carefully especially from the point of view of the (repeatable) deposition of R6G. The physisorption or chemisorption of R6G should be specified.

Author reply: Thanks for the valuable comment. A new paragraph describing the repeatable deposition of R6G during cycles of coloring/decoring or photocatalytic treatment has been included in the methods of Supporting Information. The physisorption or chemisorption of R6G has also been specified in the revised manuscript.

“Electrochromic renewing of tungsten oxide SERS substrates. Typically, following extensive washing with distilled water and/or ethanol, the adsorbed R6G molecules are further detached from the tungsten oxide substrate after SERS measurement in an aqueous electrolyte solution by applying a reverse bias of 0.2 V for 180 s. Subsequently, the electrochromic-treated substrate is re-colored to the same degree as the pristine one under a voltage of -0.5 V after again washing with distilled water and/or ethanol. Then, the reactivated substrate could be re-used for the next SERS measurement after re-loading of R6G analyte. This procedure can be repeated many times as necessary. Notably, the R6G molecules are believed to interact with our electrochromic substrates by both physisorption and chemisorption, since some part of R6G are easily detached when immersed into ethanol while some part of R6G are in no way washed away by ethanol or water. Expectedly, the chemisorption of R6G occurs at disordered regions, particularly defects on tungsten oxide substrate (i.e. oxygen vacancies, W^{5+}) by electrostatic interaction.

Photocatalytic renewing of tungsten oxide SERS substrates. Typically, following extensive washing with distilled water and/or ethanol, the R6G-adsorbed substrates are illuminated for at least 2 hours by a simulated solar source equipped with a 300 W Xenon lamp (AM 1.5). Then, the reactivated substrate could be re-used for the next SERS measurement after again washing with distilled water and/or ethanol and re-loading of R6G analyte.”

Comment 16:

16) I am not sure, if the term “regeneration” is correct. It looks like cycles, but what is regenerated. How the surface is cleaned and how the cleaning quality is evaluated?

Author reply: Thanks for the valuable comment. In biology, regeneration is the process of renewal, restoration, and growth that makes genomes, cells, organisms, and ecosystems resilient to natural fluctuations or events that cause disturbance or damage. In our system, it is found that the tungsten oxide SERS substrates after electrochromic treatment can acquire their initial SERS activities as the pristine substrates, so we believe the SERS substrates have been regenerated similar to the biological system.

In the revised manuscript, the cleaning quality has been carefully evaluated by two other means (UV-Vis and Raman spectroscopies) allowing us to validate the electrochromic technique. The relevant results have been included in the revised manuscript, as also described below.

“The efficacy of our electrochromic renewability scheme is also examined by two other means (UV-Vis and Raman spectroscopies) (Supplementary Fig. 17). It could be observed that the spectroscopic profiles obtained from the regenerated SERS substrate after electrochromic coloration are almost identical to those obtained from the freshly prepared substrate, along with the absence of clearly discerned signals ascribed to R6G analyte, indicating good renewability that makes the surface able to recover.”

Comment 17:

17) Row 444 The DOS abbreviation should be explained in the text (not only in the figure caption).

Author reply: Thanks for the valuable comments. The DOS abbreviation has been explained

in the text as “The density of states (DOS)”.

Comment 18:

18) Row 485 The term “colorimetric enhancement” is rather misleading; “colorimetry” is an analytical technique.

Author reply: Thanks for the valuable comments. It has been corrected to “color-dependent enhancement”.

Comment 19:

19) Rows 553 – 555 What is the analyte? See the sentence “Colored films as SERS substrates were obtained after discharging potentiostatically for 180 s at certain potentials in an aqueous analyte of 1 M of chloride salt. Is it in the presence of R6G in the form of chloride? It looks peculiarly. Please, specify the chloride salt.

Author reply: Thanks for the valuable comment. Also, we are sorry for the inappropriate statement of “in an aqueous analyte of 1 M of chloride salt”. In fact, R6G molecules are not involved during the discharging process, and the chloride salt is specified as AlCl_3 in the revised manuscript.

“Colored films as SERS substrates were obtained after discharging potentiostatically for 180 s at certain potentials in 1 M aqueous AlCl_3 solution.”

Comment 20:

20) The Figure S11 is described as “The absorbance of the pristine and colored tungsten oxide films monitored at 532 nm.” However, the horizontal axis of the graph is wavelength, i.e. it is not a monitoring of absorbance at one value of wavelength.

Author reply: Thanks for the valuable comment. It has been corrected as follows: “The absorbance spectra of the pristine and colored tungsten oxide films in the 300-800 nm wavelength range.”

Comment 21:

21) The Figures S12 b is denoted as “(b) EF values calculated from the intensities ...”, the

vertical axis in the figure is described as Intensity.

Author reply: Thanks for the valuable comment. It has been corrected as follows: “(b) The Raman intensities of vibrational band at 612 cm⁻¹ of R6G on the substrates intercalated with varied cations.”

Comment 22:

22) The Figure S15 The axis description is “Subatraste number”. Please, correct typing.

Author reply: Thanks for the valuable comment. It has been corrected to “Substrate number”.

Comment 23:

23) Bulk R6G crystals on bare Si/SiO₂ wafer should be described more precisely considering their use for EF calculations.

Author reply: We greatly appreciate the valuable suggestion on EF calculations. Raman profile of bulk R6G crystals on bare Si/SiO₂ wafer has been examined, with a brief discussion included in the revised manuscript as follows:

Calculation of the enhancement factor.

The enhancement factor EF was calculated according to the formula:

$$EF = (I_{\text{SERS}}/N_{\text{SERS}})/(I_{\text{bulk}}/N_{\text{bulk}}) \quad (1)$$

$$N_{\text{bulk}} = \rho h A_{\text{Raman}} N_A / M \quad (2)$$

$$N_{\text{SERS}} = CV N_A A_{\text{Raman}} / A_{\text{sub}} \quad (3)$$

N_{SERS} and N_{bulk} denote the number of R6G molecules that contribute to the signal intensity, enhanced and normal, respectively, while I_{SERS} and I_{bulk} denote the corresponding enhanced and normal Raman intensities (equation 1). As normal reference, the data for bulk R6G crystals on bare Si/SiO₂ wafer were acquired. h is the confocal depth of the laser beam, and on the basis of molecular weight (M) and density (ρ) of bulk R6G (1.15 g cm⁻³), N_{bulk} is calculated by equation 2. For analyte molecules loaded with SERS-active substrates, N_{SERS} can be estimated by equation 3, assuming that the analyte was distributed uniformly on the surface of substrates. C is the molar concentration of the analyte solution, V is the volume of

the droplet, N_A is Avogadro constant. A_{Raman} is the laser spot area (1 μm in diameter) of Raman scanning. 50 microliters of the droplet on the substrate was spread over the surface of the film (1.5 cm \times 2.5 cm) spontaneously, from which the effective area of the substrate, A_{Sub} , can be obtained.

For the confocal depth (h) of the laser beam, recent references give different values for solid bulk crystalline R6G from 10, 13, 21 to 26 μm (Nanoscale, 2013, 5(7): 2784-2789.; J Phys. Chem. C, 2012, 116(5): 3320-3328.; Surf. Sci., 1998, 406(1-3): 9-22.; Anal. Chem., 2017, 89(21): 11765-11771.), which might be related to different pinhole sizes and objective lens of their used Raman instruments. To provide a more accurate estimate of the confocal depth in our system, a Raman intensity-depth profile of the 520.6 cm^{-1} band for a silicon wafer has been made in the light of the model assumed by Cai et al. (Surf. Sci., 1998, 406(1-3): 9-22.). Judging from the profile, the confocal depth (h) is determined to be 23.64 μm in our system (Figure S27).

Reviewer 2:

The authors introduce electrochromic WO_3 substrates as efficient and renewable SERS substrates. Here the advantage of the electrochromic behaviour is that the change in absorbance can be related to the SERS enhancement, and this can be used for enhancement calibration.

I have some general, and also few detailed questions:

Author reply: We are grateful to the “reviewer 2” for the positive and encouraging comments. Also, we have tried our best to clarify the issues concerned by the reviewers in the revised manuscript.

Comment 1:

How is the SERS enhancement factor calculated in detail? The authors should show the reference spectrum, and give the values for the calculation.

Author reply: We greatly appreciate the valuable suggestion on EF calculations.

The calculation of the enhancement factor (EF) in our work is detailed as follows:

Calculation of the enhancement factor.

The enhancement factor EF was calculated according to the formula:

$$EF = (I_{\text{SERS}}/N_{\text{SERS}})/(I_{\text{bulk}}/N_{\text{bulk}}) \quad (1)$$

$$N_{\text{bulk}} = \rho h A_{\text{Raman}} N_{\text{A}} / M \quad (2)$$

$$N_{\text{SERS}} = CV N_{\text{A}} A_{\text{Raman}} / A_{\text{sub}} \quad (3)$$

N_{SERS} and N_{bulk} denote the number of R6G molecules that contribute to the signal intensity, enhanced and normal, respectively, while I_{SERS} and I_{bulk} denote the corresponding enhanced and normal Raman intensities (equation 1). As normal reference, the data for bulk R6G crystals on bare Si/SiO₂ wafer were acquired. h is the confocal depth of the laser beam, and on the basis of molecular weight (M) and density (ρ) of bulk R6G (1.15 g cm⁻³), N_{bulk} is calculated by equation 2. For analyte molecules loaded with SERS-active substrates, N_{SERS} can be estimated by equation 3, assuming that the analyte was distributed uniformly on the surface of substrates. C is the molar concentration of the analyte solution, V is the volume of the droplet, N_{A} is Avogadro constant. A_{Raman} is the laser spot area (1 μm in diameter) of Raman scanning. 50 microliters of the droplet on the substrate was spread over the surface of the film (1.5 cm \times 2.5 cm) spontaneously, from which the effective area of the substrate, A_{sub} , can be obtained.

For the confocal depth (h) of the laser beam, recent references give different values for solid bulk crystalline R6G from 10, 13, 21 to 26 μm (Nanoscale, 2013, 5(7): 2784-2789.; J Phys. Chem. C, 2012, 116(5): 3320-3328.; Surf. Sci., 1998, 406(1-3): 9-22.; Anal. Chem., 2017, 89(21): 11765-11771.), which might be related to different pinhole sizes and objective lens of their used Raman instruments. To provide a more accurate estimate of the confocal depth in our system, a Raman intensity-depth profile of the 520.6 cm⁻¹ band for a silicon wafer has been made in the light of the model assumed by Cai et al. (Surf. Sci., 1998, 406(1-3): 9-22.). Judging from the profile, the confocal depth (h) is determined to be 23.64 μm in our system (Figure S27).

The relevant information has been included in the revised manuscript (Supplementary Methods).

Comment 2:

The elemental maps in my documents are essentially black, I cannot discern anything there.

Author reply: Thanks for the valuable comment. The blurry elemental mapping images have been replaced in the revised manuscript.

Comment 3:

How does the regeneration work exactly? Are the analyse molecules etched away during the electrochemical process? To prove this, the authors should show the spectra before and after regeneration, which means with the molecules at a given concentration, and then after the electrochromic regeneration of the “clean”, but already intercalated sample. This would show that no trace amounts of the previous analyse remain on the substrate.

Author reply: Thanks for the valuable comment. The regeneration work has been more detailly described as follows:

Electrochromic renewing of tungsten oxide SERS substrates:

Typically, following extensive washing with distilled water and/or ethanol, the adsorbed R6G molecules are further detached from the tungsten oxide substrate after SERS measurement in an aqueous electrolyte solution by applying a reverse bias of 0.2 V for 180 s. Subsequently, the electrochromic-treated substrates are re-colored to the same degree as the pristine one under a voltage of -0.5 V after again washing with distilled water and/or ethanol. Then, the reactivated substrate could be re-used for the next SERS measurement after re-loading of R6G analyte. This procedure can be repeated many times as necessary. Notably,

the R6G molecules are believed to interact with our electrochromic substrates by both physisorption and chemisorption, since some part of R6G are easily detached when immersed into ethanol while some part of R6G are in no way washed away by ethanol or water. Expectedly, the chemisorption of R6G occurs at disordered regions, particularly defects on tungsten oxide substrate (i.e. oxygen vacancies, W^{5+}) by electrostatic interaction.

Photocatalytic renewing of tungsten oxide SERS substrates:

Typically, following extensive washing with distilled water and/or ethanol, the R6G-adsorbed substrates are illuminated for at least 2 hours by a simulated solar source equipped with a 300 W Xenon lamp (AM 1.5). Then, the reactivated substrate could be re-used for the next SERS measurement after again washing with distilled water and/or ethanol and re-loading of R6G analyte.

To examine the efficacy of our electrochromic renewability scheme, each step of the renewability or purification has been carefully monitored by two other means (UV-Vis and Raman spectroscopies). The relevant results have been included in the revised manuscript, as also described below.

“The efficacy of our electrochromic renewability scheme is also examined by two other means (UV-Vis and Raman spectroscopies) (Supplementary Fig. 17). It could be observed that the spectroscopic profiles obtained from the regenerated SERS substrate after electrochromic coloration are almost identical to the those obtained from the freshly prepared substrate, along with the absence of clearly discerned signals ascribed to R6G analyte, indicating good renewability that makes the surface able to recover.”

Figure S17. (a) UV-Visible absorption spectra and (b) Raman spectra at 532 nm laser

excitation of a series of substrates including the pristine tungsten oxide film, the tungsten oxide film loaded with R6G analytes and the regenerated tungsten oxide films with different regeneration cycles by means of electrochromic treatments.

Comment 4:

It would be nice to see that the SERS enhancement works also with other molecules than R6G.

Author reply: Thanks for the valuable suggestion. As suggested by the reviewer, in addition to R6G, other types of molecules, such as crystal violet (CV) and Victoria blue B (VBB), have also been selected as analytes for SERS detection with our electrochromic SERS substrates. Distinguishable SERS signals related to CV (10^{-3} M) and VBB (10^{-3} M) can be observed on the electrochromic SERS substrates.

Figure S11. SERS spectra of (a) crystal violet (10^{-3} M) and (b) Victoria blue B (10^{-3} M) on the pristine (bleached) and Li-intercalated tungsten (colored) oxide films under 532 nm laser excitation. A comparison of SERS activity between the pristine (bleached) and Li-intercalated (colored) tungsten oxide film under 532 nm laser excitation, for (c) vibrational band at 913

cm^{-1} of CV and (d) vibrational band at 1617 cm^{-1} of VBB. Cation intercalation is conducted in 1 M aqueous LiCl analyte solution via chronoamperometry at constant potential of -0.5 V for 180 s.

Comment 5:

What are the restrictions of SERS with semiconductor substrates? Does the excitation have to be above the band gap of the molecule? What are the resonance conditions? Would it also work at 785 nm?

Author reply: We greatly appreciate for the valuable comments.

- (1) About the restrictions. It should be emphasized that the band level alignment between the semiconductor substrate and the analyte is crucial to the semiconductor SERS based on the theory of chemical mechanism (CM). In fact, only when a good band matching between the semiconductor substrate and the analyte is achieved, the resonance conditions for several resonance modes can be fulfilled in the constructed semiconductor/analyte system, including the molecule resonance, the exciton (or interband) transition resonance, and the photon-induced charge-transfer resonance together with the ground-state charge-transfer interactions. Accordingly, only effective resonances would lead to a magnification of Raman scattering cross-section and then strong SERS enhancements.
- (2) About the demand for band-gap structures. Excitations above the band gaps of analyte molecules are not necessary on semiconductor SERS enhancement. As a matter of fact, if the charge transfer from the semiconductor substrate to the analyte or vice versa could be manipulated to get close to the energy of the incident laser, strong SERS enhancement is also likely to occur. However, the molecular transition between the HOMO and LUMO levels of analyte molecule may provide another resonant pathway to further enhance the SERS effect, if it is energetically near that of the excitation laser.
- (3) About the resonance conditions. The existence of charge-transfer transition with energy at or near that of incident laser for an effective resonance may be one of the premises for strong SERS enhancements based on semiconductors, according to the chemical mechanism (CM). However, other thermodynamically feasible resonances such as exciton transition within semiconductors and molecular transition of analyte have also been found

to play crucial roles in the SERS enhancement of semiconductors through vibronic coupling, according to the theory established by Lombardi et al (Chem. Rev., 2016, 116(24): 14921-14981). On coupling, the relatively weak charge-transfer resonance borrows intensity from the stronger nearby resonances, which can be expressed by Herzberg-Teller coupling terms for intensity borrowing from molecular and interband transitions, respectively. In addition, the ground-state charge-transfer (GSCT) effect may also be expected for the semiconductor/analyte system, which may act as a possible contributor to the prominent SERS enhancement.

- (4) About working at 785 nm. SERS detection of analyte molecules using 633 or 785 nm laser excitation is also possible, only if the resonance conditions are fulfilled in the constructed semiconductor/analyte systems. However, the SERS signals are usually much weaker as compared to the wavelength of 532 or 633 nm in our experiments, since the excitation wavelength (785 nm) is not in direct resonance with the absorptions of most analyte molecules.

Figure S12. (a) SERS spectra of R6G (10^{-4} M) on the pristine (bleached) and Li-intercalated tungsten oxide films (colored) under 633 nm laser excitation. A comparison of SERS activity between the pristine (bleached) and Li-intercalated (colored) tungsten oxide film under 633 nm laser excitation for vibrational band at 612 cm^{-1} of R6G. Cation intercalation is conducted in 1 M aqueous LiCl analyte solution via chronoamperometry at constant potential of -0.5 V for 180 s.

Comment 6:

How are the samples treated after the electrochemical intercalation? Are they washed somehow, or just left to dry? This should be described in detail. In Figure 1c is the absorbance recorded in solution? And it would be nicer to have color photos in the insets of 1c.

Author reply: Thanks for the valuable suggestion.

(1) Experimental details of preparation and characterization of the samples treated after the electrochemical intercalation has been described in the revised manuscript.

“Typically, SERS spectra of R6G molecules deposited on the pristine and colored films as substrates were obtained under laser excitation at 532.8 nm, after the films were thoroughly rinsed with distilled water, dried at 60 °C and re-loaded with R6G.”

(2) Yes, the absorbance in Figure 1c is recorded in solution. The caption of Figure 1c has been changed to “in 1M AlCl₃ aqueous solution”

(3) The corresponding high-resolution color photos have been provided in the Supporting Information (Figure S6).

Figure S6. (a) Electrochromic switching, optical absorbance monitored at 532 nm for the sputtered tungsten oxide film, with alternating bias potentials switching between -0.5 and 0.2 V in 1 M AlCl₃ aqueous solution. High-resolution photos of the tungsten oxide film in its bleached (b) and colored (c) states.

Comment 7:

How can the effect that more SERS effective substrate can be distinguished from less effective one by the naked eye be useful? After all, it is the Raman spectrum from the molecule that is of interest, and the substrate color is not sensitive to the signal from the molecule.

Author reply: Thanks for the valuable suggestion. The useful feature has been emphasized in the supporting Information (Supplementary Note S4) in the revised manuscript as follows:

“About the usefulness of colorimetric functionality.

As is well known, commercially available metallic SERS substrates based on well-organized ‘hotspot’ structures often have the quality guarantee period emphasized in their user’s manual. In fact, their performances such as activity and reproducibility will be gradually deteriorated once unpacked for uses, since their delicate surface structuring is easily destroyed. Unfortunately, the performance degradation for metallic SERS substrates cannot be easily perceived by the naked eyes, because the quality-degraded substrates have the same appearances as the fresh one, which may cause users great trouble in accurately analyzing of SERS signals. In contrast, the electrochromic SERS substrates with inherent colorimetric functionality have the ability to reveal their SERS activity through a color change observable by the naked eye, which provide a visual, rapid determination about the SERS-active status of SERS substrates.”

Comment 8:

Does Figure 1b show 50 curves that fall exactly onto each other? The caption suggests that.

Author reply: Thanks for the valuable comment.

Yes, the 50 curves overlap virtually perfectly in the Figure, implying good reversibility between the coloring and bleaching states. The information has been added in the Figure caption.

Comment 9:

An AFM image at the same magnification as the SEM image in Fig 1a should be supplied to get insight into the grains of the film.

Author reply: Thanks for the valuable comment. As suggested, an AFM image at the same magnification as the SEM image in Fig 1a has been included in the revised manuscript.

Figure S1. AFM measurements show low variation in the surface roughness for (a) the sputtered tungsten oxide film, and relatively large surface roughness for (b) the tungsten oxide film prepared by drop casting method.

Comment 10:

In Figure 2b I find it daring to speak of a plateau at -1.2. Then, why do the lower 3 data points do not have error bars? What is the goodness of a linear fit to that data set?

Author reply: We greatly appreciate the valuable comments.

(1) The inappropriate description of “plateau” has been replaced by the word “maximum” in the revised manuscript.

“Impressively, when plotted versus the absorbance, the EF is found to increase almost linearly up to 2.66×10^4 , reaching a maximum within the absorbance range of 0.3-1.2, nearly 28 times greater than that of the pristine uncolored film”

(2) Error bars has been added for the lower 3 data points.

(3) The coefficient of determination (R^2) has been given as a judgement of the goodness of a linear regression to the data set, which is calculated to be 0.96 within the absorbance

range of 0.3-1.2.

Comment 11:

The comparative figures 3b and S13 should have values at the y axis, such that absolute signal versus deviation becomes evident.

Author reply: Thanks for the valuable comment. The values at the y axis have been included in Figure 3b and Figure S13.

Comment 12:

It would be better to give the absorbance in meaningful units such as optical density and not in a.u.

Author reply: Thanks for the valuable comment. By definition, the absorbance is the common logarithm of the ratio of incident to transmitted radiant power through a material, so the result of absorbance is dimensionless. In the revised manuscript, we have removed the units (a. u.) of the absorbance measured by UV-Vis spectra, according to previously published works (*Chem. Eng. J.* 323 (2017) 37, *Talanta* 167 (2017) 310, *J. Phys. Chem. C* 121 (2017) 16481, *Phys. Rev. B* 97 (2018) 201414(R), *J. Phys. D: Appl. Phys.* 50 (2017) 39LT01).

Comment 13:

Substrate description jumps between ITO and FTO

Author reply: Thanks for the valuable comment. We are sorry for such a simple mistake, and the word “ITO” has been corrected to “FTO” in the revised manuscript.

Comment 14:

On page 13 they say “only few studies have reported truly renewable SERS substrates”, these should be cited here.

Author reply: We greatly appreciate the suggestions on the references. Some of relative references have been cited in the revised manuscript.

Comment 15:

Pages should be numbered.

Author reply: We greatly appreciate the valuable suggestion. Page number has been included in the revised manuscript.

Comment 16:

The paper can be written in a more concise fashion, introduction is a bit repetitive, with I do not know how many times “uniformity, reproducibility, and renewability”.

Author reply: We greatly appreciate the valuable suggestion. We have rewritten the introduction part in a more concise fashion, just as suggested by the reviewer.

For example:

“Such a chemical interaction-based mechanism offers new opportunities for the simultaneous acquisition of uniformity, reproducibility, and renewability, together with Raman signal amplification using semiconductor SERS substrates.” has been changed to “Such a chemical interaction-based mechanism offers new opportunities for the simultaneous fulfillment of important criteria for consideration in SERS measurements using semiconductor SERS substrates.”

“More importantly, a clear quantitative relationship can be found between the SERS enhancement of the colored substrate and the amount of intercalated charges, which can be adjusted by varying the negative voltage. This enables a controlled modulation of the chemical and electronic structures of semiconductor SERS substrates, leading to the realization of uniformity, reproducibility, and renewability.” has been changed to “More importantly, a clear quantitative relationship can be found between the SERS enhancement of the colored substrate and the amount of intercalated charges by systematically varying the negative voltage, which enables a controlled modulation of the chemical and electronic structures of semiconductor SERS substrates and in turn affects their SERS performances.

Comment 17:

Also, when the sample description starts, first 6 supporting Figure are discussed before the first main text figure. For the reader it would be nicer to come more straight to the point.

Author reply: We greatly appreciate the valuable suggestions. In the revised manuscript, part

of the figures about the pristine uncolored substrates in the Supporting Information has been included in the main text, as depicted in Figure 1.

Comment 18:

To my opinion, the manuscript needs a major revision with substantial additional experimental data before it can be reconsidered for publication in Nat. Commun.

Author reply:

Thanks for the valuable comment. As suggested by the reviewers, we have reconstructed the manuscript and revised it substantially with new experimental data as well as relative discussions added. In the revised manuscript, blurred figures have been replaced, and experimental details have been described. Additionally, the results of the electrochromic substrates working at other excitation wavelengths (633 nm) and the SERS enhancement towards other molecules (crystal violet and Victoria blue B) have also been examined.

Reviewers' Comments:

Reviewer #1:

Remarks to the Author:

I appreciate the effort of authors to improve the manuscript and to accept the recommendations of reviewers.

I would like to say that new data were supplemented and the text was improved substantially.

I recommend accepting the manuscript.

Reviewer #2:

Remarks to the Author:

he authors have supplied sufficient additional data and overall have satisfied my comments.

The manuscript could still profit from language improvements.

COMMENTS BY REVIEWERS:

Reviewer #1

I appreciate the effort of authors to improve the manuscript and to accept the recommendations of reviewers. I would like to say that new data were supplemented and the text was improved substantially. I recommend accepting the manuscript.

Author reply: Thanks for the kind comments of the reviewer.

Reviewer #2

The authors have supplied sufficient additional data and overall have satisfied my comments. The manuscript could still profit from language improvements.

Author reply: Thanks for the kind comments of the reviewer. We have tried our best to further improve the language of the manuscript in this final revision, with shortened and polished context as requested.